# A Simple Framework for Generalization in Visual RL under Dynamic Scene Perturbations

**Wonil Song**
Yonsei University
Seoul, South Korea
swonil92@yonsei.ac.kr

**Hyesong Choi**
Ewha Womans University
Seoul, South Korea
hyesongchoi2010@gmail.com

**Kwanghoon Sohn**
Yonsei University
Seoul, South Korea
khsohn@yonsei.ac.kr

**Dongbo Min**[*]
Ewha Womans University
Seoul, South Korea
dbmin@ewha.ac.kr

## Abstract

In the rapidly evolving domain of vision-based deep reinforcement learning (RL), a pivotal challenge is to achieve generalization capability to dynamic environmental changes reflected in visual observations. Our work delves into the intricacies of this problem, identifying two key issues that appear in previous approaches for visual RL generalization: (i) imbalanced saliency and (ii) observational overfitting. Imbalanced saliency is a phenomenon where an RL agent disproportionately identifies salient features across consecutive frames in a frame stack. Observational overfitting occurs when the agent focuses on certain background regions rather than task-relevant objects. To address these challenges, we present a simple yet effective framework for generalization in visual RL (SimGRL) under dynamic scene perturbations. First, to mitigate the imbalanced saliency problem, we introduce an architectural modification to the image encoder to stack frames at the feature level rather than the image level. Simultaneously, to alleviate the observational overfitting problem, we propose a novel technique called shifted random overlay augmentation, which is specifically designed to learn robust representations capable of effectively handling dynamic visual scenes. Extensive experiments demonstrate the superior generalization capability of SimGRL, achieving state-of-the-art performance in benchmarks including the DeepMind Control Suite. [1]

## 1 Introduction

Deep reinforcement learning (RL) utilizing visual observations has achieved remarkable success across diverse domains, including robotic manipulation [21], video games [25, 1, 38], and autonomous navigation [24, 48]. However, acquiring generalizable RL policies across diverse environments remains challenging, mainly due to overfitting [44] in the high-dimensional observation space [32]. To obtain robust policies invariant to visual perturbations, a variety of approaches based on domain randomization [36, 28] and data augmentation [31, 19, 40, 29, 13, 12] have been widely proposed. These approaches operate under the assumption that exposing an agent to various augmentations during the training phase enhances its adaptability to unseen domains. Despite the encouraging results, performance still falls behind in challenging environments with dynamically changing backgrounds.

---

[*]Corresponding Author
[1]Website and code are available at: https://w-song11.github.io/SimGRL.

38th Conference on Neural Information Processing Systems (NeurIPS 2024).

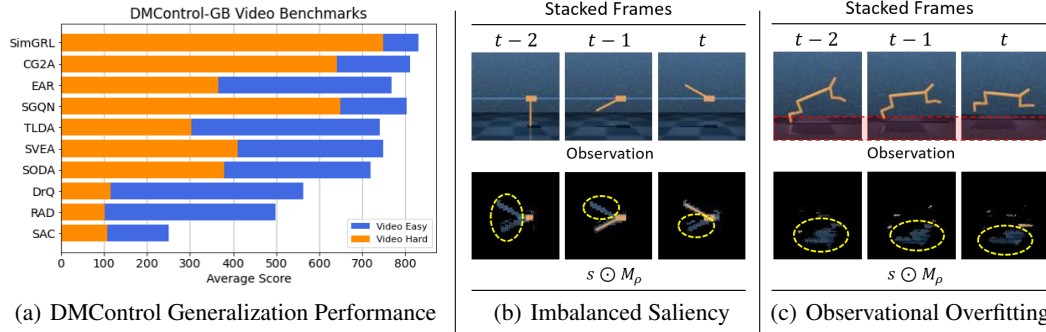

| (a) DMControl Generalization Performance | (b) Imbalanced Saliency | (c) Observational Overfitting |

Figure 1: (a) Average performances on 6 tasks in DMControl-GB. In contrast to other methods with significant performance degradation in Video Hard, our proposed SimGRL demonstrates robust performance across all benchmarks. (b)−(c) Examples of two problematic phenomena that can cause overfitting in visual RL generalization. The background structures in the red boxes are correlated with the movement of the task object. $s$ and $M_\rho$ represent the stacked frames and attribution masks, respectively. Attribution masks in this figure were obtained using the critic trained by DrQ [40].

Fig. 1(a) shows the notable performance degradation of existing approaches when comparing 'Video Hard' scenes with dynamically changing backgrounds to less dynamic 'Video Easy' scenes in the DeepMind Control Suite-Generalization Benchmark (DMControl-GB) [13].

Using a gradient-based attribution mask $M_\rho$ [2], we first investigate the causes of the degradation in generalization in such challenging environments by examining salient regions across consecutive stacked frames used as an RL input. Based on our analysis, we empirically identified two phenomena, highlighting them as key causes of performance degradation: (i) what we refer to as *imbalanced saliency* and (ii) *observational overfitting* [32]. In the DMControl [35, 37], Fig. 1(b) depicts an example of the imbalanced saliency in a 'Cartpole, Swingup' task, where salient features across all stacked frames are biased to the regions corresponding to the task objects in the latest two frames. Fig. 1(c) shows an example of the observational overfitting that occurs in a 'Cheetah, Run' task. In this case, the RL agent misidentifies the ground as more salient than the 'Cheetah' object. These two problems contribute to overfitting to the training environment, making generalization of the RL agent even more challenging.

In image-based RL, the effectiveness of regularization has been demonstrated in numerous studies. DrQ [40] and RAD [19], each employing random shift and random crop augmentations as data regularization on input images, achieved remarkable performance improvements over vanilla SAC [8]. Furthermore, [32] exhibited that regularization effects from architectural modifications such as overparameterization or residual connections can help avoid observational overfitting and reduce the generalization gap.

In this context of research, we introduce a **Sim**ple yet effective framework for **G**eneralization in visual **RL** (SimGRL) under dynamic scene perturbations. Firstly, we empirically found the image-level frame stack, commonly used in traditional vision-based *model-free* RL approaches [25, 18, 41, 40, 19], to be the main factor causing the imbalanced saliency problem. To address this issue, we propose a simple architectural modification of an image encoder, which employs a feature-level frame stack instead of the image level. This involves extracting individual feature maps for each frame from the shallow layers of the encoder, stacking them along channels, and encoding the stacked feature maps through the remaining layers. Since this approach considers solely individual frames during the initial encoding, the agent is trained to focus on spatially salient features in each consecutive frame that are essential for a given task, alleviating the imbalanced saliency problem of Fig. 1(b). Secondly, to address the observational overfitting problem by encouraging the agent to focus on the task object rather than backgrounds, we propose a new data augmentation called shifted random overlay, which is a modified version of random overlay augmentation [13]. This augmentation directly injects background dynamics irrelevant to the given task into the agent during the training phase by interpolating natural images moving in a random direction on each frame. By enabling the agent to implicitly learn to ignore task-irrelevant backgrounds and focus on task-relevant pixels, this approach alleviates the observational overfitting problem of Fig. 1(c). Furthermore, this augmentation enables the agent to adapt to test environments with dynamic perturbations in the surrounding backgrounds.

We verify that these strategies can remarkably improve generalization in challenging environments, achieving state-of-the-art performance without using any additional auxiliary losses or networks.

Furthermore, utilizing the attribution mask $M_\rho$ [2], we introduce novel metrics, called **T**ask-**Id**entification (TID) metrics, consisting of TID score and TID variance, to quantitatively evaluate the discrimination ability on salient regions. With the proposed metrics, we quantitatively analyze the two problematic phenomena in the existing approaches, demonstrating the excellent discrimination ability of SimGRL. Moreover, we show a tendency of positive correlation between the TID score and RL generalization performance, emphasizing the importance of accurately identifying salient features in input images.

Our contributions include the following aspects:

- By utilizing gradient-based attribution masks, we highlight the two core issues of imbalanced saliency and observational overfitting, which hinder the generalization of visual RL for most model-free RL settings. Additionally, we propose TID metrics to measure the discrimination ability of an RL agent on task objects, providing insights into these issues.

- To address these problems, we propose architectural and data regularization methods through a modification to an encoder structure and an introduction of new data augmentation.

- We achieve state-of-the-art performances across video benchmarks of DMControl-GB [13], DistractingCS [34], and robotic manipulation tasks [14].

## 2 Background

**Visual RL and Generalization**  We consider a partially observable Markov decision process (POMDP) problem $\mathcal{M} = (\mathcal{S}, \mathcal{O}, \mathcal{A}, \mathcal{P}, r, \gamma)$, where $\mathcal{S}$ is state space, $\mathcal{O}$ is observation space, $\mathcal{A}$ is action space, $\mathcal{P} : \mathcal{S} \times \mathcal{A} \rightarrow \mathcal{S}$ is the transition function that defines the conditional probability distribution $\mathcal{P}(s_{t+1}|s_t, a_t)$ over next states given a state $s_t \in \mathcal{S}$ and an action $a_t \in \mathcal{A}$ taken at time $t$, $r : \mathcal{S} \times \mathcal{A} \rightarrow \mathbb{R}$ is a reward function, and $\gamma \in [0, 1)$ is a discount factor. In the POMDP problem, such as the visual RL, because only the high-dimensional observations $o_t \in \mathcal{O}$ can be observable [15], a state is defined as a sequence of the $k$ consecutive image frames $s_t = (o_{t-k+1}, ..., o_{t-1}, o_t)$ [25]. Without loss of generality, we will set $k = 3$ for the sake of notational simplicity. RL aims to learn a policy $\pi : \mathcal{S} \rightarrow \mathcal{A}$ that maximizes the expected sum of discounted rewards $\mathbb{E}_\pi[\sum_{t=0}^{T} \gamma^t r(s_t, r_t)]$. In this work, we focus on the generalization problem to POMDPs $\widehat{\mathcal{M}} = (\widehat{\mathcal{S}}, \widehat{\mathcal{O}}, \mathcal{A}, \mathcal{P}, r, \gamma)$, where the states $\hat{s} \in \widehat{\mathcal{S}}$ are constructed from the perturbed observations $\hat{o} \in \widehat{\mathcal{O}}$, and a POMDP $\widehat{M}$ is sampled from the space of POMDPs $\mathbb{M}$, $\widehat{M} \sim \mathbb{M}$.

**Deep Q-Learning and Soft Actor-Critic**  Deep Q-learning [25] is a common model-free RL algorithm that aims to learn a parameterized state-action value function $Q_\theta(s_t, a_t)$, where $\theta$ is the deep neural network parameters of the Q-function and action is greedily selected as the one with the maximum value $a_t = \text{argmax}_a Q_\theta(s_t, a)$ at time $t$. The training of the Q-function is achieved by minimizing a mean squared error of the Bellman residuals:

$$\mathbb{E}_{(s_t, a_t, s_{t+1}) \sim \mathcal{B}}[(Q_\theta(s_t, a_t) - (r_t + \gamma \max_{a'} Q_{\hat{\theta}}^{tgt}(s_{t+1}, a')))^2], \quad (1)$$

where $\hat{\theta}$ is the parameters of the target Q network $Q^{tgt}$ and $\mathcal{B}$ is a replay buffer. To increase the stability, the parameter of the target Q network is slowly updated by the exponential moving average (EMA) $\hat{\theta} = \lambda\theta + (1 - \lambda)\hat{\theta}$ with $\lambda \ll 1$ [22]. For the continuous action space, rather than the greedy sampling, a parameterized actor $\pi_\phi(s_t)$ is employed as the policy, where $\phi$ is the neural network parameters of the actor. Soft Actor-Critic (SAC) [8] is a common actor-critic algorithm with a state-action value $Q_\theta(s_t, a_t)$ and a stochastic policy $\pi_\phi(a_t|s_t)$, and a temperature parameter $\alpha$, which aims to optimize a $\gamma$-discounted maximum-entropy objective [49].

In visual RL, a parameterized encoder $f_\theta : \mathbb{R}^{C \times H \times W} \rightarrow \mathbb{R}^d$ is employed to compress the high-dimensional image inputs and shared by both the actor and critic, where $d$ is a dimension of the encoded feature. Consistent with the prior visual RL approaches [41, 18, 19, 40, 12], we jointly train the encoder with the critic and freeze it during actor updates, where the encoder learns representations for RL tasks by the critic loss. For notational simplicity, we denote all parameters updated by the critic loss as $\theta$ and the actor parameters as $\phi$.

**Gradient-based Attribution Mask**    An attribution map, also known as a saliency map, is designed to visualize the salient pixels in input images for given tasks. A common approach to computing the attribution map is a gradient-based method [30, 3], which indicates how sensitive the task prediction is to perturbations in the input pixels. Consistent with SGQN [2], we employ guided backpropagation [33] to compute the attribution map $M(Q_\theta, s_t, a_t) = \frac{\partial Q_\theta(s_t, a_t)}{\partial s_t}$, where $s_t$ and $M(Q_\theta, s_t, a_t) \in \mathbb{R}^{C \times H \times W}$. Then, the binarized attribution mask $M_\rho(Q_\theta, s_t, a_t)$, referred to as $\rho$-quantile attribution mask, is computed by thresholding $M(Q_\theta, s_t, a_t)$ by the $\rho$-quantile, where $M_\rho(Q_\theta, s_t, a_t)_{(i,j,k)} = 1$ if the pixel value of $M(Q_\theta, s_t, a_t)_{(i,j,k)}$ belongs to the $\rho$-quantile of highest values for $M(Q_\theta, s_t, a_t)$, and 0 otherwise. We use this attribution mask to investigate the cause of the degradation of the generalization performance of existing methods in challenging environments such as Video Hard [13]. Furthermore, to justify our analysis, we will introduce new metrics that quantitatively measure the ability to identify salient pixels in Section 4.4. Note that we leverage the attribution mask only for analysis without involving it in the training process.

## 3    Pitfalls within Conventional Practices in Visual RL Generalization

The generalization ability of RL agents is often degenerated in challenging environments characterized by dynamic perturbations and different structures compared to the training environment. In this section, we investigate the causes of the overfitting to the training environment observed in existing visual RL methods on the DMControl Generalization Benchmark (DMControl-GB) [13].

Conventional practices used to train the visual RL agent for generalization [13, 2, 42, 4] include frame stack and data augmentation. First, to reflect the temporal structure of the input state, Q-value is predicted using stacked frames [25, 18, 41, 40, 19] instead of a single image as follows:

$$q_\theta(s_t, a_t) = Q_\theta(f_\theta([o_{t-2}, o_{t-1}, o_t]), a_t), \quad s_t = (o_{t-2}, o_{t-1}, o_t), \tag{2}$$

where $\theta$ is a neural network parameter, $f_\theta(\cdot)$ is a convolutional neural network (CNN) image encoder, $Q_\theta(\cdot, \cdot)$ is a critic head, and $[\cdot]$ is a concatenation operator along a channel dimension. Subsequently, to learn the representations robust to visual perturbations, data augmentation to the visual observations is leveraged when training the encoder. Specifically, $s_t$ in Eq. (2) can be substituted with $\tau(s_t) = (\tau(o_{t-2}), \tau(o_{t-1}), \tau(o_t)), \tau \sim \mathcal{T}$, where $\tau(\cdot)$ is a sampled transformation function from the transformation space $\mathcal{T}$. However, we empirically found that these practices can cause the saliency imbalance between the stacked frames and easily fall into observational overfitting, resulting in performance degradation in unseen challenging environments.

**Pitfall 1: Imbalanced Saliency**    Fig. 2(a) illustrates an example of the imbalanced saliency by SVEA [12] in the 'Cartpole' task. As described in Section 2, the attribution masks were acquired by thresholding the gradient value of the critic function with respect to input image frames [2]. In this example, an RL agent recognizes the regions of the salient objects in the two latest frames as having high saliency across all stacked frames. This makes the agent misidentify as salient pixels of $o_{t-2}$ its background parts corresponding to the positions of the 'Cartpole' object in $o_{t-1}$ and $o_t$. As a result, $o_{t-2}$ provides the agent with redundant information that is unnecessary for the task, leading to overfitting to training data [45, 5]. In a test environment with complicated backgrounds, the background information that is considered salient from $o_{t-2}$ may act as noise that interferes with the decision-making process of the policy. As the latest frames are more closely related to the subsequent decision-making process, the agent tends to identify spatially consistent saliency maps based on these recent frames, leading to imbalanced saliency maps. We hypothesize that this phenomenon occurs because the encoder extracts features throughout concatenated images along the channel as in Eq. (2), forcing it to capture the same spatial saliency for all stacked images.

**Pitfall 2: Observational Overfitting**    Observational overfitting [32] can arise when certain background elements move in synchronization with task-relevant objects. It is prone to occur particularly when the motion of the task object is not substantial, such as in the 'Cheetah, Run' task. Although data augmentation is leveraged during training, we found that observational overfitting can still occur. Furthermore, as the same augmentation is applied uniformly to all consecutive frames, it does not effectively prevent the agent from erroneously focusing on background elements. For instance, Fig. 2(b) shows an example of observational overfitting that occurs in the 'Cheetah, Run' task by SVEA, where the critic assigns more significant saliency to the ground than to the 'Cheetah' object. In a test

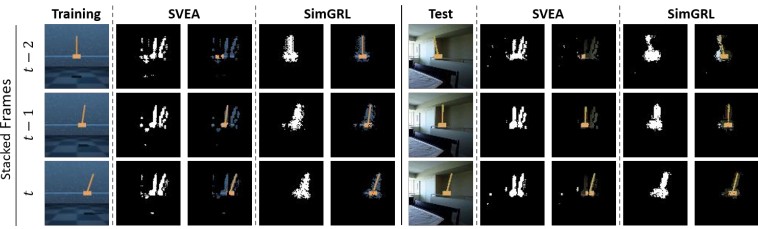

(a) Attribution masking examples in 'Cartpole, Swingup'

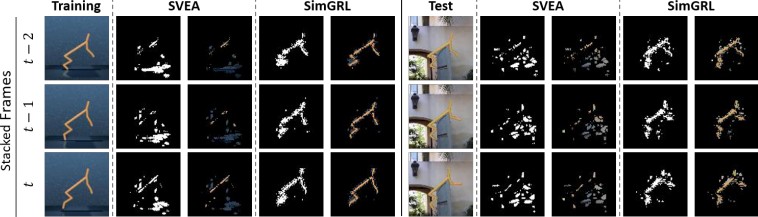

(b) Attribution masking examples in 'Cheetah, Run'

Figure 2: Examples of attribution masks and masked frames. Compared to SVEA that falls into the imbalanced saliency and observation overfitting in the 'Cartpole, Swingup' and 'Cheetah, Run' tasks, respectively, the proposed SimGRL accurately identifies the true salient pixels even in challenging 'Video Hard' test environments of DMControl-GB. We provide further examples of the attribution masks and masked salient regions for various environments and algorithms in Appendix F.4.

environment, if there is a lack of correlation between objects and backgrounds present in the training environment, the agent may struggle to make accurate decisions, thereby leading to challenges in generalization.

## 4   Method

To address the problems within the conventional practices used for the generalization of visual RL, we propose a **Sim**ple framework for **G**eneralization in visual **RL** (SimGRL) under dynamic scene perturbations.

### 4.1   Feature-Level Frame Stack

In Section 3, we identified the encoder structure that simultaneously encodes the stacked frames at the image level as the cause of the imbalanced saliency. We address this issue by introducing an architectural regularization strategy to slightly modify the encoder structure. This architectural modification involves encoding each frame individually and then encoding the stacked feature maps. To keep the computational cost almost unchanged, we partition the original encoder into two segments instead of adding new layers. Then, Eq. (2) can be modified as follows:

$$q_\theta(s_t, a_t) = Q_\theta(f_\theta^2([f_\theta^1(o_{t-2}), f_\theta^1(o_{t-1}), f_\theta^1(o_t)]), a_t), \quad s_t = (o_{t-2}, o_{t-1}, o_t), \tag{3}$$

where $f_\theta^1(\cdot)$ is an image encoder to encode individual frames, and $f_\theta^2(\cdot)$ is a feature encoder for the stacked feature maps. While $f_\theta^2$ encodes both spatial and temporal structures from inputs, $f_\theta^1$ conducts only spatial encoding of each frame. This simple modification allows the critic to be implicitly trained to focus on the spatially salient pixels of individual frames, enabling the agent to distinctly identify the salient pixels of each frame. In our experiments, we will verify that this simple modification of the encoder is highly beneficial for generalization, especially in challenging test environments.

### 4.2   Shifted Random Overlay Augmentation

The random overlay augmentation [13] used in existing approaches augments input images through linear interpolation with a natural image $\varepsilon \in \mathbb{R}^{C \times H \times W}$ randomly sampled from the Places [46] dataset that contains 1.8M diverse scenes:

$$\tau^{RO}(s; \varepsilon) = (\alpha\varepsilon + (1 - \alpha)o_1, ..., \alpha\varepsilon + (1 - \alpha)o_n), \quad \varepsilon \sim \mathcal{D}, \tag{4}$$

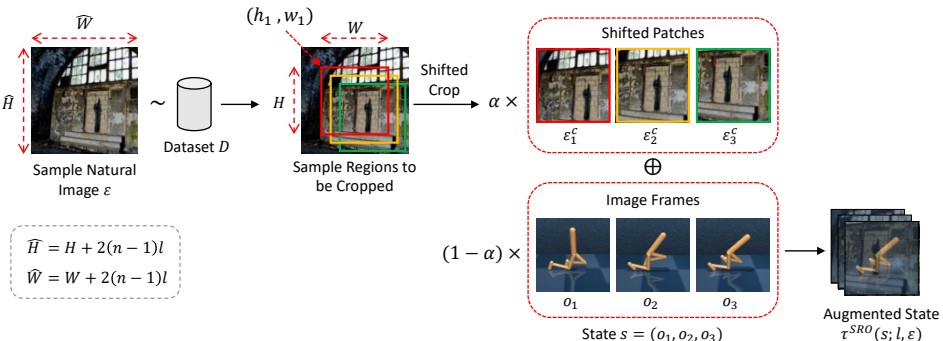

Figure 3: Shifted random overlay (SRO) augmentation for data regularization. To inject random dynamics into the backgrounds of RL input images, we generate multiple cropped patches in a shifted manner from a sampled natural image and interpolate them to augment the input images.

where $s$ is a state that consists of a sequence of images $(o_1, ..., o_n)$, $o_i \in \mathbb{R}^{C \times H \times W}$, $\mathcal{D}$ is a dataset and a common choice for the interpolation coefficient is $\alpha = 0.5$. The original random overlay augmentation uses the same natural image $\varepsilon$ for all stacked frames. Contrarily, we introduce a shifted random overlay (SRO) augmentation to inject the task-irrelevant dynamics into the training images, which is depicted in Fig. 3. Considering a maximum shift length $l$ for each shifting and a stacked frame number $n$, this method first samples a natural image $\varepsilon \in \mathbb{R}^{C \times \widehat{H} \times \widehat{W}}$ from the dataset $\mathcal{D}$, where $(\widehat{H}, \widehat{W}) = (H + 2(n-1)l, W + 2(n-1)l)$. Then, we crop $n$ shifted patches $\varepsilon_i^c \in \mathbb{R}^{C \times H \times W}$ from $\varepsilon$ to augment each frame in a shifted way. Given an upper left corner coordinate $(h_1, w_1)$ of $H \times W$ sized center crop at the center of $\varepsilon$, the coordinates $(h_i, w_i)$ of the upper left corner of $\varepsilon_i^c$ are selected as follows:

$$(h_i, w_i) = (h_1 + dh(i-1), w_1 + dw(i-1)), \qquad i = 1, ..., n, \qquad (5)$$

where $dh, dw \sim \mathrm{unif}\{-l, l\}$. Finally, using the cropped images $\varepsilon_i^c$ from $(h_i, w_i)$ in $\varepsilon$, the shifted random overlay augmentation is defined as:

$$\tau^{SRO}(s; l, \varepsilon) = (\alpha \varepsilon_1^c + (1-\alpha)o_1, ..., \alpha \varepsilon_n^c + (1-\alpha)o_n), \qquad (6)$$

where we adopt $\alpha = 0.5$. This augmentation provides two implicit advantages when training RL agents. Firstly, the inclusion of background elements in motion, independent of the task, enables the RL agent to perceive that rewards are solely associated with changes in the genuine task object. Consequently, the agent is trained to concentrate on the task object, disregarding the movement of background elements, thereby alleviating the issue of observational overfitting. Secondly, this augmentation method generates training data akin to real environments with dynamic backgrounds, enabling the agent to be robust in such conditions and enhancing generalization capability.

### 4.3 Simple Framework for Generalization in Visual RL (SimGRL)

Built on the SVEA [12] framework that leverages the data-mixing strategy between weak and strong data augmentations and computes the target Q-value using clean images for stability in SAC, we propose a Simple framework for Generalization in visual RL (SimGRL) under dynamic scene perturbations, which integrates the proposed two strategies. For the strong augmentation, we utilize the shifted random overlay instead of the original random overlay without shifting while the random shift [40] is employed as the weak augmentation. Considering that the clean images $s_t$ and $s_{t+1}$ already have weak augmentation, *i.e.*, random shift, applied by default, the critic loss for SimGRL is defined as follows:

$$\mathcal{L}_Q(\theta) = \mathbb{E}_{(s_t, a_t, r_t, s_{t+1}) \sim \mathcal{B}}[\beta(q_\theta(s_t, a_t) - q^{tgt})^2 + (1-\beta)(q_\theta(\tau^{SRO}(s_t; l, \varepsilon), a_t) - q^{tgt})^2], \quad (7)$$

where $q_\theta(\cdot, \cdot)$ is computed by Eq. (3), $q^{tgt} = r_t + \gamma q_{\hat{\theta}}(s_{t+1}, a')$, $a' \sim \pi_\phi(\cdot | f_\theta(s_{t+1}))$, and $\hat{\theta}$ is the parameter of the target network that is updated by the exponential moving average (EMA). The strong augmentation $\tau^{SRO}(\cdot; \cdot, \cdot)$ is the shifted random overlay in Eq. (6). Consistent with the SVEA, we adopt the data-mixing coefficient $\beta = 0.5$. When training the actor $\pi_\phi$, we leverage solely clean images like the existing methods [12, 13]. The overall framework of SimGRL is illustrated in Fig. 4 and the algorithm is summarized in Appendix C.

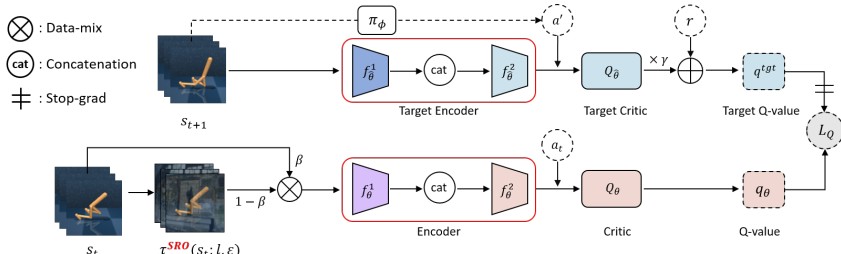

Figure 4: Overview of the **Sim**ple framework for **G**eneralization in visual **RL** (SimGRL) under dynamic scene perturbations. Differences from SVEA are marked in red.

### 4.4  **T**ask-**I**dentification (TID) Metrics

To quantitatively evaluate the capability to identify task-relevant objects in each stacked frame as salient, we introduce novel metrics, referred to as **T**ask-**I**dentification (TID) score and variance, based on the $\rho$-quantile attribution mask $M_\rho(Q_\theta, s_t, a_t)$ [2] described in Section 2.

**TID Score**   TID score measures how much the model identifies the task object's pixels across stacked frames, which is defined as:

$$TID_S = \sqrt{\frac{N_{obj_M}}{N_{obj}} \times \frac{N_{obj_M}}{N_M}} = \sqrt{\frac{(N_{obj_M})^2}{N_{obj} \times N_M}}, \tag{8}$$

where $N_{obj}$ is the number of task object's pixels in input images, $N_M$ is the number of pixels in attribution masks $M_\rho$, $N_{obj_M}$ is the number of task object's pixels included in $M_\rho$, and $\rho$ is a quantile value for thresholding the attribution map. Note that all numbers consisting of the TID score are counted across the full consecutive frames. In Eq. (8), the first term quantifies the model's ability to identify the pixels of the task object, while the second term quantifies how accurately the model identifies the task object's pixels. These two terms trade-off depending on the size of the quantile value and are upper-bounded by 1, thus leading to an upper-bounded TID score by 1. With a model that perfectly identifies all task pixels, this upper-limit value can only be achieved along with an optimal $\rho$ value, which is the quantile of the number of the task object's pixels in frames. This $\rho$ value can be computed by $\rho = 1 - \frac{N_{obj}}{(n \times C \times H \times W)}$, where $n$ is the number of frame stack. We explain the impact of the $\rho$ value on this TID score in Appendix F.2.

**TID Variance**   TID variance measures how discriminatively the model distinguishes the task object's pixels in each frame, which is defined as:

$$TID_{Var} = Var[100 \times (TID_S^1, TID_S^2, ..., TID_S^n)], \tag{9}$$

where $TID_S^i = \sqrt{\frac{(N_{obj_M}^i)^2}{N_{obj}^i \times N_M^i}}$, $N_{obj}^i$, $N_M^i$, and $N_{obj_M}^i$ are individually counted at each frame. Since each $TID_S^i$ is upper-bounded by 1, the values of $TID_{Var}$ become too small for meaningful comparison. Therefore, we multiply $TID_S^i$ by 100 to obtain a variance with a more appropriate scale for comparison.

## 5  Experiments

In this section, we present the experimental results of SimGRL on the DMControl-GB [13] video benchmarks ('Video Easy' and 'Video Hard') at 500K simulated training frames to demonstrate the generalization capability under dynamic scene perturbations. To tackle the vision-based control tasks with continuous action space, we employ the SAC [8] as a backbone RL algorithm, and compare SimGRL against current state-of-the-art

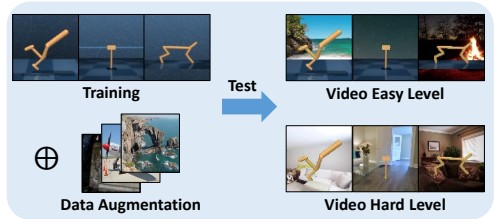

Figure 5: Experimental setup. We evaluated the zero-shot performances for test environments with dynamic background perturbations.

Table 1: Generalization performance on video benchmarks from DMControl-GB [13]. We report the results on mean and standard deviation over 5 seeds. The scores of the comparison methods were taken from their respective papers, and in cases where the scores were unavailable, they were obtained through our implementation using the official codes. Δ indicates the difference with second best.

| | DMControl-GB | SAC | RAD | DrQ | SODA | SVEA | TLDA | SGQN | EAR | CG2A | SimGRL | Δ |
|---|---|---|---|---|---|---|---|---|---|---|---|---|
| **Video Easy** | Walker, Walk | 245±165 | 608±92 | 747±21 | 768±38 | 819±71 | 868±63 | 910±24 | 913±38 | **918±20** | 910±21 | -8 (0.8%) |
| | Walker, Stand | 389±131 | 879±64 | 926±30 | 955±13 | 961±8 | 973±6 | 955±9 | 970±23 | 968±6 | **973±4** | 0 |
| | Ball In Cup, Catch | 192±157 | 363±158 | 380±188 | 875±56 | 871±106 | 855±56 | 950±24 | 911±40 | 963±28 | **964±7** | +1 (0.1%) |
| | Cartpole, Swingup | 398±60 | 473±54 | 459±81 | 758±62 | 782±27 | 671±57 | 761±28 | 762±88 | 788±24 | **838±35** | +50 (6%) |
| | Finger, Spin | 206±169 | 516±113 | 599±62 | 695±97 | 808±33 | 744±18 | 956±28 | 717±51 | 912±69 | **983±2** | +27 (3%) |
| | Cheetah, Run | 73±18 | 153±7 | 270±16 | 268±10 | 251±17 | **336±57** | 289±35 | 334±56 | 314±49 | 317±16 | -19 (6%) |
| **Video Hard** | Walker, Walk | 122±47 | 80±10 | 121±52 | 312±32 | 385±63 | 292±133 | 739±21 | 383±59 | 687±18 | **773±31** | +34 (5%) |
| | Walker, Stand | 231±57 | 229±45 | 252±57 | 771±83 | 834±46 | 595±56 | 851±24 | 744±62 | 895±35 | **932±17** | +37 (5%) |
| | Ball In Cup, Catch | 101±37 | 98±40 | 100±40 | 327±100 | 403±174 | 304±58 | 782±57 | 320±48 | 806±44 | **902±19** | +96 (12%) |
| | Cartpole, Swingup | 158±17 | 152±29 | 136±29 | 429±64 | 393±45 | 308±44 | 544±43 | 375±37 | 472±24 | **727±23** | +183 (34%) |
| | Finger, Spin | 13±10 | 39±20 | 38±13 | 302±41 | 335±58 | 256±25 | 822±24 | 277±62 | 819±38 | **864±12** | +42 (5%) |
| | Cheetah, Run | 75±14 | 21±9 | 49±13 | 130±24 | 112±12 | 67±23 | 157±69 | 91±46 | 168±16 | **301±7** | +133 (79%) |

methods for visual RL generalization including RAD [19], DrQ [40], SODA [13], SVEA [12], TLDA [42], SGQN [2], EAR [4], and CG2A [23]. For weak augmentation, RAD and SODA utilize random crop, while the others employ random shift, except for SAC, which does not use any data augmentations. In addition, all competitors leverage the original random overlay augmentation without shifting for strong augmentation except for RAD and DrQ, which use only weak augmentations. As default for SimGRL, we used the random shift augmentation [40] and denoted the images with only this weak augmentation as clean images with the data distribution of the training data. Additionally, we employed the proposed shifted random overlay (SRO) for strong augmentation in SimGRL and denoted images together with this augmentation as augmented ones. Implementation details are described in Appendix A, and further experimental results on other benchmarks of Distracting Control Suite [34] and robotic manipulation [14] are provided in Appendices B.7 and B.8, respectively.

## 5.1 Results on DMControl-GB

We evaluated the zero-shot generalization performance on 6 tasks in 'Video Easy' and 'Video Hard' benchmarks from DMControl-GB [13]. As illustrated in Fig. 5, the easy version shares certain structures with the training environment such as the ground and shadow, while the hard version shares nothing other than the agent's object by replacing all backgrounds with distracting videos. In Table 1, SimGRL demonstrates state-of-the-art performance in 4 out of 6 tasks in the Video Easy benchmark and achieves comparable performance in the remaining two tasks, including 'Walker, Walk' and 'Cheetah, Run'. On the other hand, our approach shows outstanding performance in all 6 tasks at the Video Hard level, where SimGRL achieves performance gain by 15% on average compared to SGQN. Specifically, SimGRL outperforms existing methods by a significant margin in 'Cartpole, Swingup' and 'Cheetah, Run', which were previously difficult to solve. We suggest that the reason for this lies in the fact that 'Cartpole, Swingup' requires detailed identification of salient objects, while 'Cheetah, Run' needs to address the observational overfitting. Both issues are mitigated in SimGRL. We provide the training curves in Appendix B.6.

**Computational Efficiency** Our method has the advantage of improving generalization performance without any extra models or auxiliary losses. We compare SimGRL with SGQN [2], the existing state-of-the-art method in the Video Hard benchmark. In experiments, SimGRL can achieve a throughput of 9.54 FPS, which is 1.55× efficient compared to SGQN's 6.16 FPS on a single NVIDIA TITAN RTX GPU. Fig. 6 shows the training curves for the zero-shot test performances over the wall-clock training time, where the averaged performances across the 6 tasks in Table 1 are presented. In this figure, SimGRL requires 44% less wall-clock training time than SGQN to reach the same 500K frames for training, demonstrating the high computational efficiency of SimGRL. It is worth noting that SimGRL has achieved considerable enhancements in both training efficiency and test

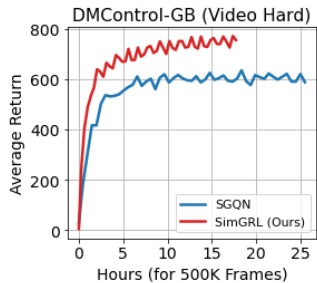

Figure 6: Training curves at wall-clock time axis.

performance compared to the SOTA method, regardless of the typical trade-offs in terms of efficiency and performance.

**Ablation Study**    To verify the effectiveness of the proposed regularizations, we compare the ablation variants with the SVEA baseline. Fig. 7 shows training curves for zero-shot test performances averaged across the 6 tasks of DMControl-GB at the Video Hard benchmark as a function of the number of stepped training frames. This result indicates that the additions of each regularization to SVEA can remarkably improve the generalization ability of the model. In particular, we emphasize that each of the two regularizations leads to sufficient improvement, where all variants of SimGRL achieve, on average, nearly 82% better performance than SVEA. Full experimental results for each task are provided in Appendix B.1 and further in-depth discussions on the effectiveness of each regularization are described in Appendix B.2.

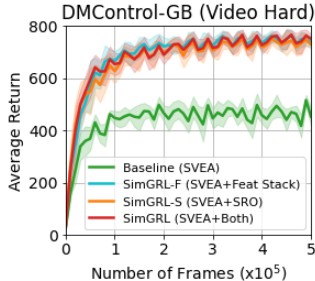

Figure 7: Training curves for ablation variants.

## 5.2    Analysis with TID Metrics

To analyze the two potential overfitting problems, namely imbalanced saliency and observational overfitting, we evaluate the TID metrics, including the TID score and variance. For clarity, we compare three methods of DrQ [40], SVEA [12], and SimGRL as the former serve as baselines for the latter. We analyze two representative tasks, namely 'Cartpole, Swingup' and 'Cheetah, Run', where each problem is prominently observed. Full evaluations for all algorithms and tasks are provided in Appendix F.3. The left plot in Fig. 8 depicts that the overall distribution of training scores of the TID metrics is divided into three regions. In this plot, both DrQ and SVEA exhibit results divided into two regions, either 'low score and low variance' or 'middle score and high variance', depending on the task, indicating that different phenomena appear in the two tasks. The 'low score and low variance' region can be interpreted as observational overfitting, which does not identify the correct object in all frames. On the other hand, the 'middle score and high variance' area suggests that the correct object is accurately recognized only in certain frames, implying imbalanced saliency. We note that although SVEA [12] incorporates a data-mixing strategy with strong augmentation to enhance the generalization capability of DrQ [40], both problems are still observed with only minor improvements over DrQ. In contrast, SimGRL reports comparatively high scores and low variances for both tasks, indicating that it can effectively identify task-relevant objects regardless of the type of task. The middle plot shows a positive correlation between training and test TID scores at Video Hard from DMControl-GB [13]. This suggests that a high discrimination ability during training can result in a high discrimination ability during testing. Finally, the right plot shows the generalization gaps, which are computed by the performance differences between training and testing, against the TID scores. This suggests that a high discrimination ability of the task object can lead to improved generalization capability.

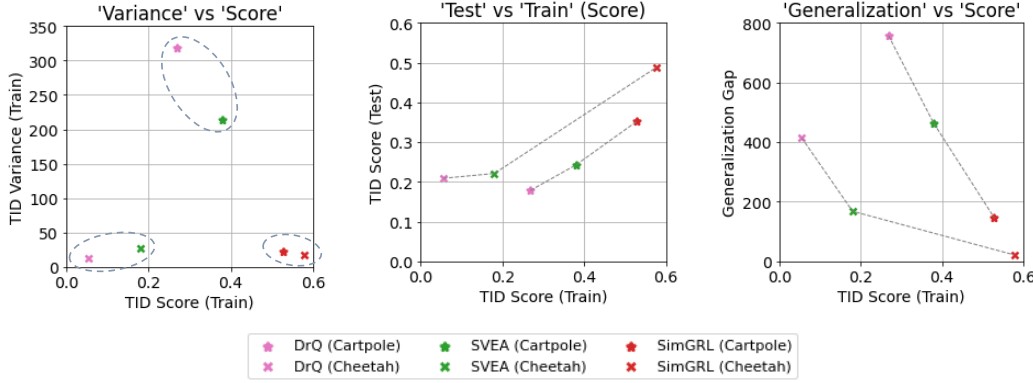

Figure 8: (Left to Right) Plots for the training TID variance, test TID scores, and generalization gap against training TID score.

## 6 Related Works

The field of domain generalization for RL has garnered considerable attention in recent years where various approaches aim to enhance the robustness of policies against visual changes. A promising approach is adapting the policy to a test domain. For example, PAD [11] suggests adopting a self-supervised task to obtain a free training signal during deployment. On the other hand, one method, proposed in [36], involves using randomly simulated RGB images. Similarly, [27] train domain-adaptive policies by randomizing dynamics during the training phase. Several works explore data augmentation techniques to improve policy generalization capacity [19, 40, 29]. For instance, RAD [19] and DrQ [40] achieve significant improvement through random crop and shift, while DrAC [29] automatically identifies the most effective augmentation with regularization terms for the policy and value function. Instead of relying solely on augmented data for policy learning, SODA [13] aims to decouple augmentation from policy learning by using soft-augmented data for policy learning and strong-augmented data for auxiliary representation learning. In a recent development, SVEA [12] designs a stabilized Q-value estimation framework to address instability issues under strong data augmentation in off-policy RL, while DBC [43] uses bisimulation metrics to learn a representation that disregards task-irrelevant information. VAI [39] extracts a universal visual foreground mask to provide an invariant observation to RL. Similarly, several methods leverage salient masks to encourage the agent to focus on import pixels. For example, TLDA [42] attempts to only augment the task-irrelevant pixels using masks obtained from the Lipschitz constant of the policy while SGQN [2] proposes saliency-guided self-supervised learning using a gradient-based saliency mask. EAR [4] attempts to learn environment-agnostic representations to enhance the robustness of policies against visual perturbations. On the other hand, several methods such as TIA [7], DRIBO [6], and RePo [47], leverage model-based RL through a recurrent state-space model (RSSM) [10, 9] to explicitly learn latent representations that focus on task-relevant features while discarding task-irrelevant ones. Our work focuses on improving the generalization ability of the RL agent based on implicit regularization approaches.

## 7 Conclusion

In this paper, we identified two key issues that degrade generalization in vision-based RL, particularly under dynamic scene perturbations, by employing a gradient-based attribution mask. To resolve these issues, we proposed a simple framework involving an architectural modification to the encoder and a new data augmentation. The proposed SimGRL algorithm has achieved state-of-the-art results on Video Hard environments in DMControl-GB, outperforming existing methods that have yet to address these challenges. Additionally, we introduced TID metrics to quantitatively evaluate the ability to discriminate task-relevant objects of RL agents. Using these metrics, we demonstrated that improving the identification of task-relevant objects can enhance the generalization capability of the vision-based RL agent.

## Acknowledgments and Disclosure of Funding

This research was supported by the National Research Foundation of Korea (NRF) grant funded by the government of Korea (MSIP) (NRF2021R1A2C2006703), by the Basic Research Lab Program through the NRF of Korea (RS-2023-00222385) and by the Yonsei Signature Research Cluster Program of 2024 (2024-22-0161).

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

# A Implementation Details

In this section, we describe the implementation details used for SimGRL. For a fair comparison, we followed the same architectural designs and hyperparameters used in [13], with only a minor difference in the encoder. For the feature-regularized encoder, we used 3 layers for the image encoder $f^1$ with 16 channels and 8 layers with 32 channels for the feature encoder $f^2$, leading to the encoder with a total of 11 layers that are the same layer number as the encoder used in [13]. Subsequently, we projected the convolutional features of the last layer to 100-dimensional linear vectors, which are fed into actor and critic heads with 1024-dimensional hidden layers. We updated the actor parameters by freezing the encoder, and the target network parameters using an exponential moving average (EMA) with a rate of 0.01 for the critic head and 0.05 for the encoder. Both actor and target networks were updated every 2 critic updates. We used a minibatch size of 128, stacked 3 image frames each of size $3 \times 84 \times 84$ as input for RL, and employed the Adam optimizer [17]. For the maximum shift length $L$ in the shifted random overlay augmentation, we employed $L = 6$, which leads to $3 \times 108 \times 108$ size of a natural image $\varepsilon$ before shifted cropping. In Fig. 9, we present examples of augmented images by the shifted random overlay. This figure exhibits that the proposed augmentation approach injects randomly moving backgrounds into the training images. All hyperparameters are summarized in Table. 2

Table 2: Hyperparameters used in DMControl Suite.

| SAC hyperparameters | Value |
|---|---|
| Replay buffer capacity | 500,000 |
| Number of training steps | 500,000 |
| Frame size | $3 \times 84 \times 84$ |
| Stacked frames | 3 |
| Evaluation episodes | 30 |
| Random shift | Up to $\pm 4$ pixels |
| Action repeat | 2 (Finger, Spin) |
| | 8 (Cartpole, Swingup) |
| | 4 (otherwise) |
| Minibatch size | 128 |
| Discount factor ($\gamma$) | 0.99 |
| Optimizer | Adam ($\beta_1 = 0.9$, $\beta_2 = 0.999$ for $\theta$ and $\phi$) |
| | Adam ($\beta_1 = 0.5$, $\beta_2 = 0.999$ for $\alpha$ of SAC) |
| Learning rate | $1e-3$ ($\theta$ and $\phi$) |
| | $1e-4$ ($\alpha$ of SAC) |
| Actor ($\phi$) update frequency | 2 |
| Target ($\hat{\theta}$) update frequency | 2 |
| Target ($\hat{\theta}$) EMA rate | 0.01 (critic) |
| | 0.05 (encoder) |
| Number of conv layers | 3 ($f^1$, Image encoder) |
| | 8 ($f^2$, Feature encoder) |
| Number of conv filters | 16 ($f^1$, Image encoder) |
| | 32 ($f^2$, Feature encoder) |
| Projection dim | 100 |
| Hidden dim (MLP heads) | 1024 |
| Initial temperature | 0.1 |
| SVEA coefficient ($\beta$) | 0.5 |
| Weak augmentation | Random shift |
| Strong augmentation | Shifted random overlay (SRO) |
| Maximum shift length ($l$) in SRO | Up to $\pm 6$ pixels |

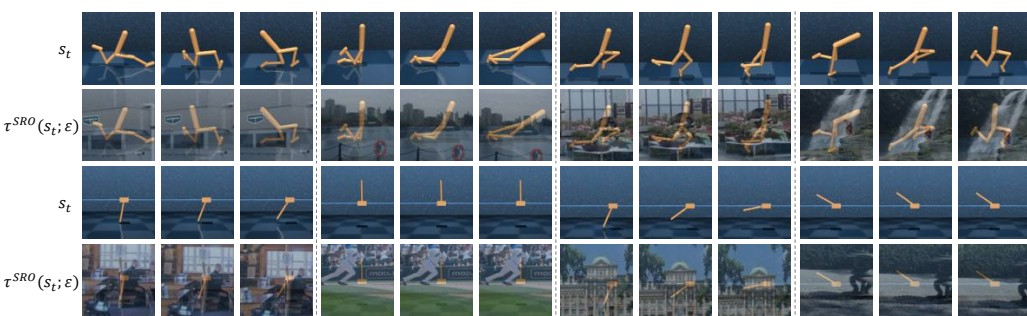

Figure 9: Examples of augmented images using shifted random overlay (SRO).

# B  Additional Experimental Results

## B.1  Detailed Results on Ablation Study

We provide detailed results of the ablation study in Table 3. For DrQ, we utilized $[K = 1, M = 1]$ [40]. In this table, by introducing weak augmentation using random shift into the vanilla SAC [8], DrQ demonstrates highly improved performance in image-based RL at the training level. While DrQ reports the best results in most tasks at the training level, the performance drastically decreases in testing levels, implying overfitting to the training environments. SVEA [12] introduces a data-mixing strategy to enhance training stability under strong data augmentation, thereby improving DrQ's generalization capability in distracting backgrounds using random overlay [13]. However, the performance of SVEA still falls behind in challenging environments such as Video Hard. To address this issue, we proposed the feature-level frame stack and shifted random overlay augmentation, resulting in significant improvements compared to the SVEA baseline in Table 3, especially in the Video Hard environments. It is noteworthy that each of the proposed regularizations demonstrates significant improvements compared to SVEA, where SimGRL-F and SimGRL-S apply the feature-level frame stack and the shifted random overlay augmentation to SVEA, respectively. Especially, all of SimGRL's variants reach the saturated performance in several tasks at the Video Hard level except for 'Walker, Walk' and 'Cheetah, Run'. For more complicated tasks such as 'Walker, Walk' and 'Cheetah, Run' tasks where performance saturation has not yet been reached, the integration of the regularizations (*i.e.*, SimGRL) clearly shows the performance improvement. Fig. 10 shows the training curves of ablations for each task at Video Hard.

Table 3: Results of the ablation study in DMCotrol-GB video benchmarks. In strong data augmentation, RO and SRO indicate the original and shifted random overlay data augmentations, respectively. Each score represents the mean and standard deviation over 5 seeds.

| | Algorithm | Frame Stack | | Weak Aug. | Strong Aug. | | Walker, Walk | Walker, Stand | Ball In Cup, Catch | Cartpole, Swingup | Finger, Spin | Cheetah, Run |
|---|---|---|---|---|---|---|---|---|---|---|---|---|
| | | Image | Feature | Random Shift | RO | SRO | | | | | | |
| **Train** | SAC | ✓ | | | | | 257±49 | 861±23 | 148±114 | 807±10 | 640±117 | 214±26 |
| | DrQ | ✓ | | ✓ | | | **943±3** | **973±2** | **975±5** | **878±2** | 982±4 | **463±11** |
| | SVEA | ✓ | | ✓ | ✓ | | 923±14 | 972±4 | 974±5 | 872±4 | 976±15 | 296±10 |
| | SimGRL-F | | ✓ | ✓ | ✓ | | 903±25 | 972±3 | 972±4 | 865±7 | 982±4 | 310±24 |
| | SimGRL-S | ✓ | | ✓ | | ✓ | 901±36 | 973±3 | 974±3 | 868±13 | 982±2 | 314±18 |
| | SimGRL | | ✓ | ✓ | | ✓ | 923±17 | 971±2 | 969±3 | 873±5 | **986±2** | 322±23 |
| **Video Easy** | SAC | ✓ | | | | | 245±165 | 389±131 | 192±157 | 398±60 | 206±169 | 73±18 |
| | DrQ | ✓ | | ✓ | | | 747±21 | 926±30 | 380±188 | 459±81 | 599±62 | 270±16 |
| | SVEA | ✓ | | ✓ | ✓ | | 819±71 | 961±8 | 871±106 | 782±27 | 808±33 | 251±17 |
| | SimGRL-F | | ✓ | ✓ | ✓ | | 873±22 | 971±4 | 947±4 | **844±15** | 975±3 | 290±37 |
| | SimGRL-S | ✓ | | ✓ | | ✓ | 878±32 | 973±5 | 962±10 | 826±25 | 962±21 | 301±23 |
| | SimGRL | | ✓ | ✓ | | ✓ | **910±21** | **973±4** | **964±7** | 838±35 | **983±2** | **317±16** |
| **Video Hard** | SAC | ✓ | | | | | 122±47 | 231±57 | 101±37 | 158±17 | 13±10 | 75±14 |
| | DrQ | ✓ | | ✓ | | | 121±52 | 252±57 | 100±40 | 136±29 | 38±13 | 49±13 |
| | SVEA | ✓ | | ✓ | ✓ | | 385±63 | 834±46 | 403±174 | 393±45 | 335±58 | 112±12 |
| | SimGRL-F | | ✓ | ✓ | ✓ | | 729±39 | 928±7 | **907±6** | 719±34 | 878±21 | 264±13 |
| | SimGRL-S | ✓ | | ✓ | | ✓ | 727±36 | 918±9 | 861±36 | 708±4 | **886±4** | 269±11 |
| | SimGRL | | ✓ | ✓ | | ✓ | **773±31** | **932±17** | 902±19 | **727±23** | 864±12 | **301±7** |

Figure 10: Trainig curves of ablation variants including the SVEA baseline at the Video Hard level.

## B.2 Discussion on Impacts of Proposed Regularizations

We discuss the impacts of the proposed regularizations based on the attribution masks and the TID metrics. First, we investigate the impacts in terms of the imbalanced saliency in the 'Cartpole, Swingup' task. In the left plot of Fig. 11, both SimGRL-F and SimGRL-S show increased TID scores compared to SVEA. However, they exhibit different effects on TID variance as SimGRL-F significantly reduces it but SimGRL-S rather increases it. This indicates that the imbalanced saliency remains in SimGRL-S while SimGRL-F alleviates this issue. Therefore, the performance improvement of SimGRL-F can be interpreted as a mitigation of the imbalanced saliency problem, while the performance improvement of SimGRL-S is a bit ambiguous. For the reason of the result with SimGRL-S, we suggest that despite the persistence of imbalanced saliency, the explicit injection of dynamic background elements might have contributed to improved generalization in such conditions. Furthermore, even though it does not distinctly discriminate task objects between the stacked frames, the increased TID score of SimGRL-S can be attributed to its improved accuracy in identifying recognized objects in certain frames, as depicted in Fig. 12. This implies that the shifted overlay augmentation implicitly encourages the model to identify task objects more accurately. As a result, SimGRL, which integrates both architectural and data regularization, demonstrates that its agent can identify task-relevant objects across stacked frames with greater accuracy and discrimination.

In the right plot of Fig. 11, all of the SimGRL variants demonstrate remarkably increased TID scores in the 'Cheetah, Run' task. Since the 'Cheetah, Run' task has less movement of task objects than the 'Cartpole, Swingup' task, differences in TID variance values for this task are not effective. Therefore, the core issue in this task is the observational overfitting [32] to ground or shadow rather than imbalanced saliency between consecutive frames. The increase in the TID score implies an enhanced ability to identify task-relevant objects, which is demonstrated in Fig. 12. As expected, this shows that the application of the shifted random overlay augmentation encourages the agent to focus on the task object while disregarding background elements. On the other hand, it is noteworthy that the frame stack at the feature level also contributes to solving this problem. We suppose this is because encoding from a single image without temporal information allows the model to concentrate on each image's most salient visual features. Consequently, by integrating those strategies, the proposed approach, SimGRL, effectively addresses the two problems highlighted in this paper that hinder generalization in visual RL.

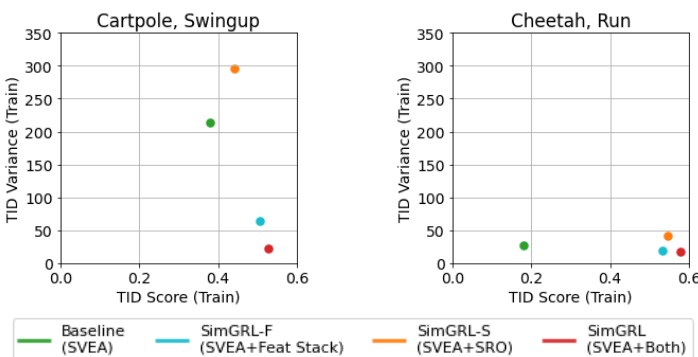

Figure 11: Plots of 'TID variance (Train) vs TID score (Train)' for ablation variants.

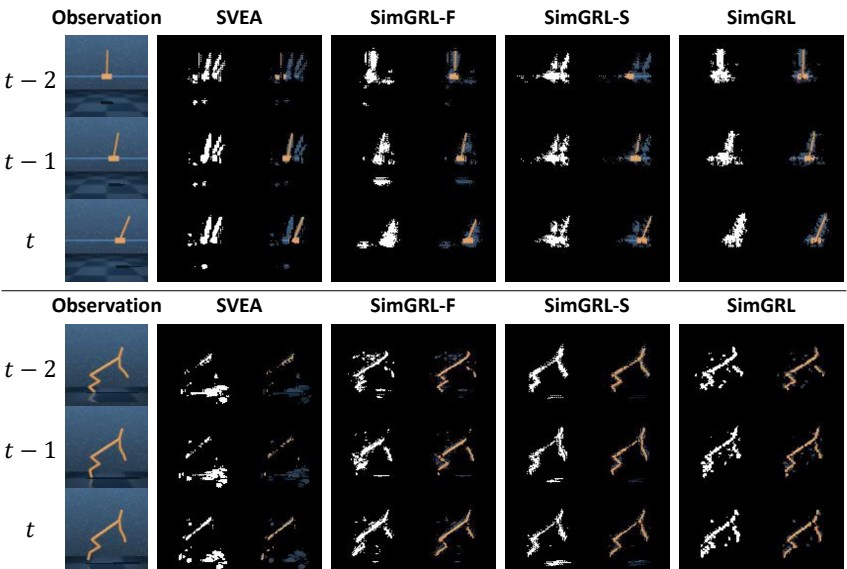

Figure 12: Examples of attribution masking for ablation variants.

## B.3 Impact of Number of Layers in Image Encoder

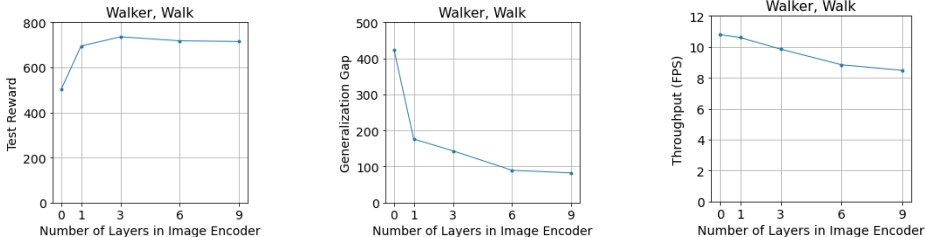

Figure 13: Impacts of the number of layers in image encoders.

While maintaining the number of layers in the entire encoder at 11, we conducted the ablation study for the number of layers in the image encoder. To consider only the impact of the layer number, we leveraged the SimGRL-F model which employs the original random overlay without shifting for the strong augmentation rather than the shifted version. Depending on the number of layers in the image encoder, we investigated test rewards, generalization gaps, and throughputs in the 'Walker, Walk' task, where the Video Hard domain was employed for testing environments. The generalization gap was evaluated by the difference between the training and testing performances, and the throughput was computed by the FPS processed in the simulator on a single NVIDIA TITAN RTX GPU. The results presented in Fig. 13 demonstrate that the incorporation of the feature-level frame stack, facilitated by the image encoder, enhances the model's generalization capabilities in dynamic scenes. It is noteworthy that even if the number of image encoder layers is only 1, the generalization gap is remarkably reduced. Additionally, the generalization gap decreases when the number of layers in the image encoder is increased, indicating that incorporating feature maps with higher abstraction enables the improvement of generalization. However, despite the improvement in generalization performance, there does not seem to be a significant increase in practical test performance, implying a reduction in the training performance through the mitigation of overfitting. This can also be attributed to the diminished depth of the feature encoder, which is crucial for encoding temporal information. Moreover, increasing the layer number in the image encoder results in decreased throughput, primarily due to the increased computational burden associated with processing individual frames. Considering both generalization capability and computational efficiency, we selected 3 layers for the image encoder and 8 layers for the feature encoder for the experiments.

## B.4 Different Data Augmentations

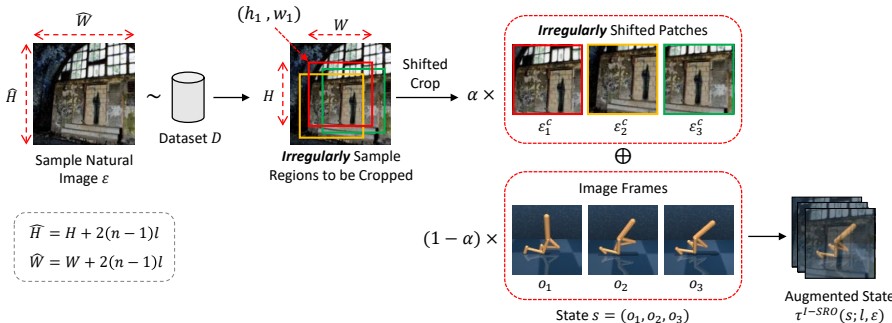

(a) Irregularly Shifted Random Overlay (I-SRO)

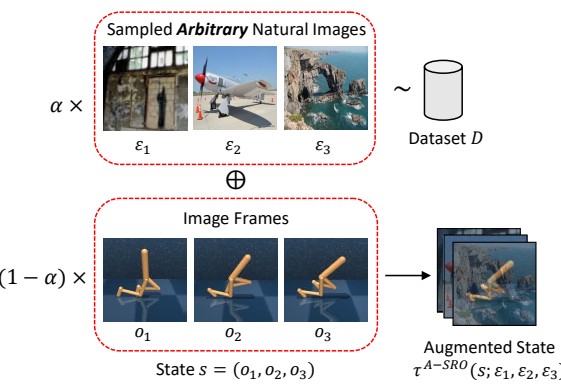

(b) Arbitrarily Stacked Random Overlay (A-SRO)

Figure 14: Additional data augmentation approaches for ablation study.

To investigate the impact of differently augmenting the stacked frames, we provide additional experiments under two types of additional augmentations: (1) randomly cropped patches from the same image and (2) completely different images. We denote the first method as *irregularly* shifted random overlay (I-SRO) and the second method as *arbitrarily stacked* random overlay (A-SRO). While the SRO augmentation crops patches in a regularly shifted manner as Eq. (5), I-SRO irregularly crops patches from a sampled natural image $\varepsilon$. In contrast to Eq. (5), the coordinates $(h_i, w_i)$ of the upper left corner of each cropped patch become as follows:

$$(h_i, w_i) = (h_1 + dh_i, w_1 + dw_i), \qquad i = 2, ..., n, \qquad (10)$$

where $dh_i, dw_i \sim \mathrm{unif}\{-l, l\}$. On the other hand, A-SRO samples different natural images $\varepsilon_i \sim \mathcal{D}$ with $H \times W$ sizes and augments each RL frame using the different images. The processes of I-SRO and A-SRO are illustrated in Fig. 14.

For clarity of the effect for each augmentation, we do not include the feature-level frame stack and compare them in the video hard level with SRO (*i.e.*, SimGRL-S in the paper) that leverages *regularly* shifted cropped patches from the same image for data augmentation. Table 4 shows that augmentations using irregularly shifted patches (I-SRO) or completely different patches (A-SRO) across the consecutive frames can achieve comparable performance with SRO using the regularly shifted patches, implying that augmenting each frame in a different manner (*i.e.*, employing SRO, I-SRO, and A-SRO) can achieve significant performance improvement over augmenting each frame uniformly (*i.e.*, RO) for visual RL generalization under dynamic scenes. Among such augmentations, we have employed the (regularly) shifted random overlay (*i.e.*, SRO) as our primary approach for data augmentation considering its superior averaged performance.

Table 4: Performance on DMControl-GB at video hard level for different strong data augmentations. The scores were evaluated over 5 seeds. Percentages indicate variations compared to RO.

| Environment | RO (=SVEA) | SRO (=SimGRL-S) | I-SRO | A-SRO |
|---|---|---|---|---|
| Walker, Walk | 385±63 | 727±36 (+89%) | 715±31 (+86%) | 732 ±25 (90%) |
| Walker ,Stand | 834±46 | 918 ±9 (+10%) | 918±8 (10%) | 910±12 (+9%) |
| Ball In Cup, Catch | 403±174 | 861±36 (+114%) | 835±42 (107%) | 832±28 (+106%) |
| Cartpole, Swingup | 393±45 | 708±4 (+80%) | 697±9 (+77%) | 705±5 (+79%) |
| Finger, Spin | 335±58 | 886±4 (+164%) | 829±20 (147%) | 804±16 (+140%) |
| Cheetah, Run | 112±12 | 269±11 (+140%) | 232±17 (+107%) | 237±21 (+111%) |

## B.5 Adoption of SGQN

Based on its architectural simplicity and absence of additional auxiliary losses beyond the actor and critic losses, we adopted the SVEA [12] algorithm as our baseline to clarify our contribution. However, our approach that utilizes two types of regularizations can be seamlessly integrated with any other algorithms in a plug-and-play manner. On the zero-shot evaluation in the video hard test environments, we present the experimental results of SGQN [2] + F (Feature-level frame stack), SGQN + S (Shifted random overlay), and SGQN + S + F in Table 5. This table shows that the proposed regularizations can also contribute to improved generalization for other methods, such as SGQN, demonstrating the generality of our method.

Table 5: Performance on DMControl-GB at video hard level for SGQN applying the proposed approaches. The scores were evaluated over 5 seeds. Percentages indicate variations compared to SGQN.

| Environment | SGQN | SGQN + F | SGQN + S | SGQN + F + S |
|---|---|---|---|---|
| Walker, Walk | 739±21 | 773±18 (+4.6%) | 786±22 (+6.4%) | 811±13 (+9.7%) |
| Walker ,Stand | 851±24 | 885±34 (+4%) | 906±15 (+6.5%) | 913 ±14 (+7.3%) |
| Ball In Cup, Catch | 782±57 | 826±24 (+5.6%) | 894±38 (14.3%) | 901±12 (+15.2%) |
| Cartpole, Swingup | 544±43 | 679±34 (+24.8%) | 632±31 (+16.2%) | 678±25 (+24.6%) |
| Finger, Spin | 822±24 | 886±22 (+7.8%) | 877±18 (+6.7%) | 881±16 (+7.2%) |
| Cheetah, Run | 157±69 | 214±52 (+36.3%) | 295±11 (+87.9%) | 311±6 (+100%) |

## B.6 Training Curves on DMControl-GB

In Figures 15, 16, and 17, we illustrate the training curves by re-implementing the comparison algorithms ourselves using available official codes over 5 seeds.

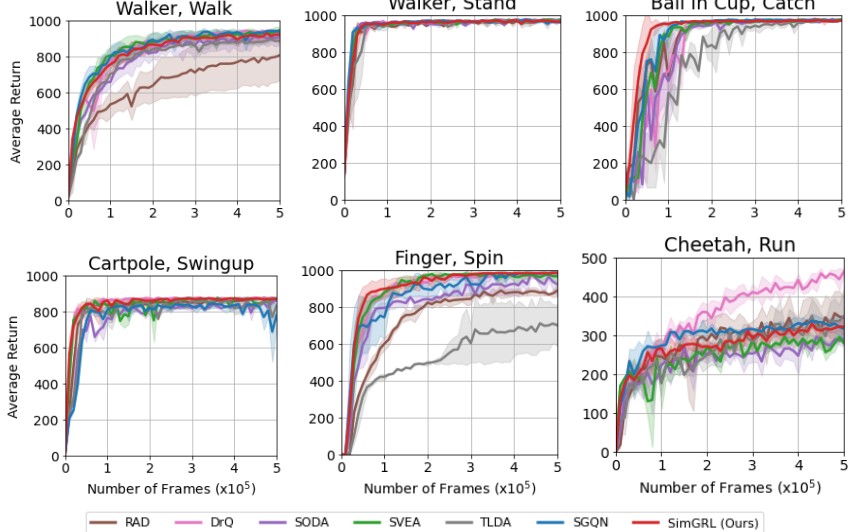

Figure 15: Training curves on the training level.

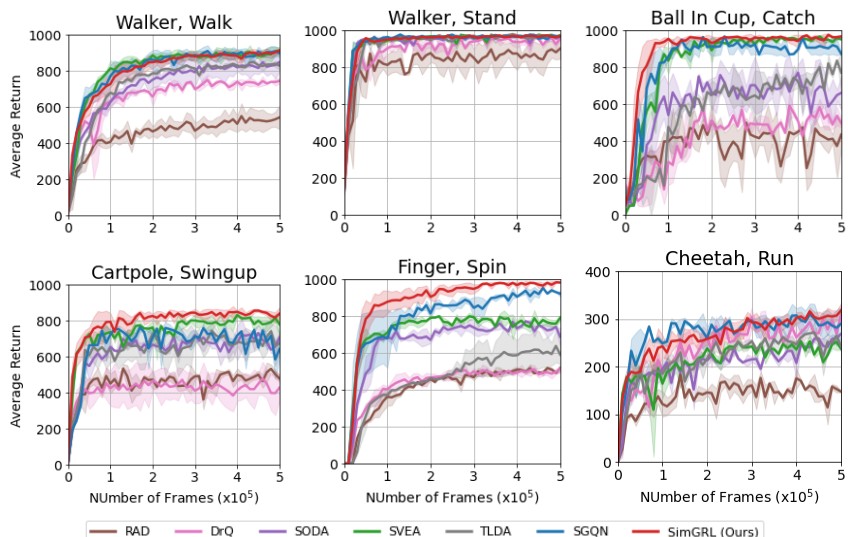

Figure 16: Training curves on the Video Easy testing level.

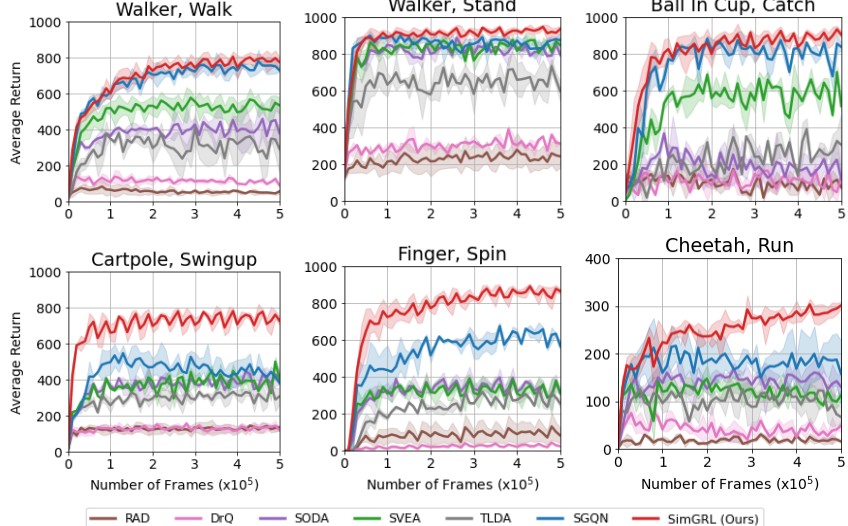

Figure 17: Training curves on the Video Hard testing level.

## B.7 Results on DistractingCS

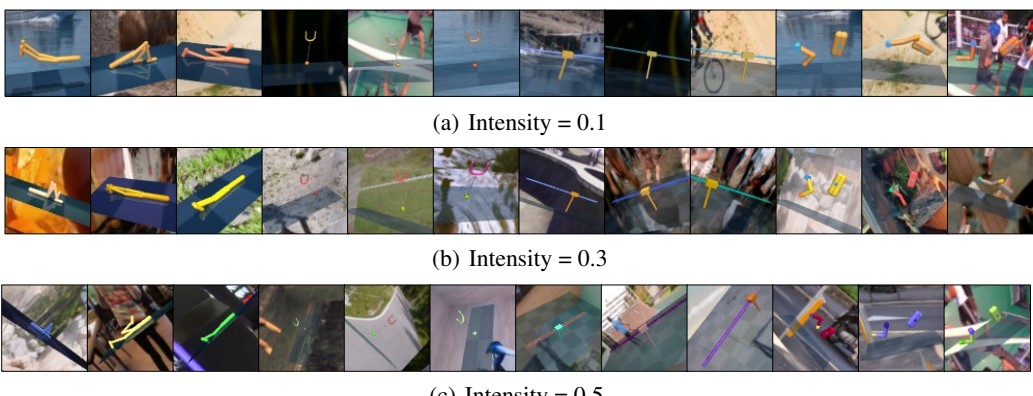

(a) Intensity = 0.1

(b) Intensity = 0.3

(c) Intensity = 0.5

Figure 18: Examples of DistractingCS benchmark.

We provide the additional benchmark results on the Distracting Control Suite (DistractingCS) [34], where camera pose, background, lighting, and colors continually vary throughout an episode. In Fig. 18, we depict some examples of the DistractingCS in intensity levels = $[0.1, 0.3, 0.5]$. In this experiment, we utilized the same hyperparameters used in DMControl-GB. Fig. 19 demonstrates SimGRL's significantly improved generalization performances in low intensities, compared to DrQ and SVEA. In particular, SimGRL is the only method that reported practical performances in the 'Ball in Cup, Catch' and 'Finger, Spin' tasks. Furthermore, SimGRL achieves additional performance gains in the 'Walker, Walk, Catch' and 'Walker, Stand' tasks. On the other hand, despite the still low performance in 'Cartpole, Swingup', there are clear signs of improvement. These results indicate the robustness of our proposed approach not only to dynamic background changes but also to various forms of distortions, including camera pose variations at moderate intensities. Finally, we note that generalization in extremely distorted environments, such as those with high intensities in the DistractingCS, remains an open problem and will be a subject of promising future research.

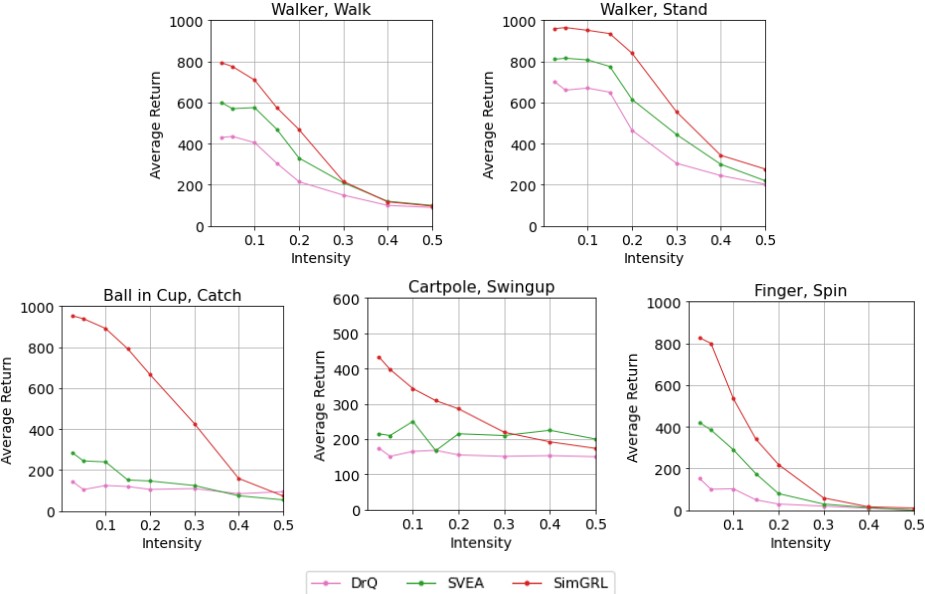

Figure 19: Performance on DistractingCS. Each score reports the mean over 5 seeds.

## B.8 Results on Robotic Manipulation

To verify the generalization capabilities of the proposed approach across various domains, we conducted additional experiments for robotic manipulation tasks [14], where the agents are trained for 250K steps. In these experiments, since the test environments introduce color and texture changes rather than structured distractions, SGQN [2] employed random convolution [20, 19] for strong augmentation. However, for SimGRL, we still utilized the shifted random overlay. In addition, for shifting, we employed two frames as an input rather than a single image setup as in SGQN. Fig. 20 exhibits examples of two robotic manipulation tasks, including 'Reach' and 'Peg in Box', and each of the testing environments. In these examples, SimGRL demonstrates its ability to appropriately capture salient regions, characterized by the robot arm and target positions. Table 6 reports the experimental results of the two tasks. Although this benchmark primarily features color and texture changes rather than dynamic background perturbations, SimGRL demonstrates superior results, achieving the best performance in 1 out of 3 tests in the 'Reach' task and in 2 out of 3 tests in the 'Peg in Box' task, and the second best in the remaining tests. It is noteworthy that SimGRL, which leverages only a variant of the random overlay, provides comparable performance to CG2A [23] that employs both random convolution and overlay augmentations.

Table 6: Performance on Robotic Manipulation. The scores were evaluated over 5 seeds.

| Task | Environment | SAC | SODA | SVEA | SGQN | CG2A | SimGRL | Δ |
|------|-------------|-----|------|------|------|------|--------|---|
| **Reach** | Train | 9.7±22 | 31.8±1 | 32.2±0 | 31.8±1 | **39.6±4** | 33.5±1 | -6.1 |
| | Test 1 | -20.9±16 | -30.9±43 | -17.6±10 | 14.4±14 | 28.7±1 | **32.6±1** | +3.9 |
| | Test 2 | -21.9±14 | -20.2±29 | -2.1±39 | 31.0±3 | **36.7±4** | 31.9±2 | -4.8 |
| | Test 3 | -43.2±6 | -68.4±30 | 1.4±29 | 29.2±7 | **35.4±4** | 33.3±1 | -2.1 |
| | Test Average | -28.6±8 | -39.9±31 | -6.1±23 | 24.9±6 | **33.6±3** | 32.6±1 | -1.0 |
| **Peg in Box** | Train | -46.7±7 | 180.0±1 | 177.5±1 | 183.9±1 | 189.9±11 | **213.6±21** | +23.7 |
| | Test 1 | -59.6±26 | 16.9±44 | -21.3±10 | -72.0±14 | 155.4±17 | **181.7±25** | +26.3 |
| | Test 2 | -60.15±10 | 0.7±30 | 96.8±40 | 110.7±3 | **157.8±22** | 138.5±18 | -19.3 |
| | Test 3 | -48.8±17 | 73.6±31 | 40.5±28 | 154.6±7 | 174.0±21 | **191.4±27** | +17.4 |
| | Test Average | -56.2±7 | 30.4±31 | 38.6±23 | 64.4±6 | 162.4±20 | **170.5±23** | +8.1 |

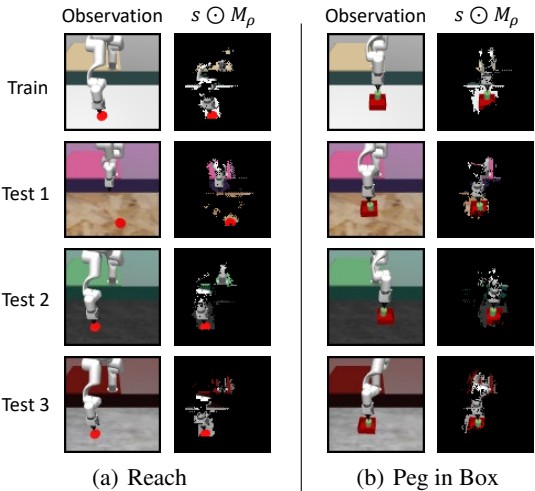

(a) Reach    (b) Peg in Box

Figure 20: Examples of Robotic Manipulation Tasks.

## C  Algorithm for SimGRL

---

**Algorithm 1** A Simple Framework for Generalization in visual RL (SimGRL).

---

1: **Hyperparameters:** Total number of training steps $T$, mini-batch size $N$, learning rate $\eta$, target network update rate $\lambda$, actor and target network update frequency $M$, transformation function set $\mathcal{V}$ for random shift, natural image dataset $\mathcal{D}$ for SRO, critic loss coefficient $\beta$, maximum shift length $l$ in SRO, replay buffer $\mathcal{B}$, discount factor $\gamma$.

2: **Initialize:** encoder and critic parameters $\theta$, actor parameters $\phi$, target network parameters $\hat{\theta} \leftarrow \theta$.

3: **for** timestep $t = 1...T$ **do**

4:    $a_t \sim \pi_\phi(\cdot|f_\theta(s_t))$        ▷ Sample action from actor (= policy)

5:    $s'_t \sim \mathcal{P}(\cdot|s_t, a_t)$        ▷ Sample transition from environment

6:    $\mathcal{B} \leftarrow \mathcal{B} \cup (s_t, a_t, r(s_t, a_t), s'_t)$      ▷ Add transition to replay buffer

7:    $\{(s_i, a_i, r(s_i, a_i), s'_i)\}_{i=1}^N \sim \mathcal{B}$      ▷ Sample batch from replay buffer

8:    **for** $i = 1...N$ **do**

9:      $s_i = \tau(s_i; \nu_i),\ s'_i = \tau(s'_i; \nu'_i),\ \nu_i, \nu'_i \sim \mathcal{V}$     ▷ Apply random shift augmentation

10:      $q_i^{tgt} = r(s_i, a_i) + \gamma Q_{\hat{\theta}}(f_{\hat{\theta}}(s'_i), a'_i),\ a'_i \sim \pi_\phi(\cdot|f_\theta(s'_i))$    ▷ Compute target Q-value

11:      $s_i^{aug} = \tau^{SRO}(s_i; l, \varepsilon_i),\ \varepsilon_i \sim \mathcal{D}$      ▷ Apply SRO augmentation

12:    **end for**

13:    $\mathcal{L}_Q(\theta) = \frac{1}{N}\sum_{i=1}^N [\beta(Q_\theta(f_\theta(s_i), a_i) - q_i^{tgt})^2 + (1-\beta)(Q_\theta(f_\theta(s_i^{aug}), a_i) - q_i^{tgt})^2]$

14:                 ▷ Compute SimGRL loss

15:    $\theta \leftarrow \theta - \eta\nabla_\theta \mathcal{L}_Q(\theta)$      ▷ Optimize encoder and critic for SimGRL loss

16:    **if** every $M$ step **then**

17:      $\phi \leftarrow \phi - \eta\nabla_\phi \mathcal{L}_\pi(\phi)$      ▷ Optimize actor for actor loss

18:      $\hat{\theta} \leftarrow \lambda\theta + (1-\lambda)\hat{\theta}$      ▷ Update target network using EMA

19:    **end if**

20: **end for**

21: Note: $f_\theta(s_t)$ represents $f_\theta^2([f_\theta^1(o_{t-2}), f_\theta^1(o_{t-1}), f_\theta^1(o_t)])$, where $s_t = (o_{t-2}, o_{t-1}, o_t)$.

---

## D  Analysis of Attentions

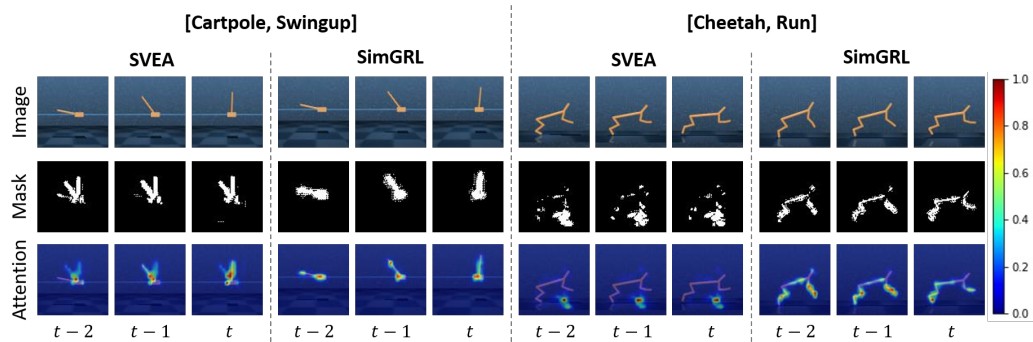

Figure 21: Attention maps were obtained from the softmax of the saliency maps before thresholding by quantile to compute the mask. The attribution masks indicate that SimGRL can effectively distinguish task-relevant objects and backgrounds, implying mitigation of the imbalanced saliency and observation overfitting. Furthermore, the attention maps indicate that SimGRL can focus on the significant parts of the task objects. In contrast, the SVEA baseline focuses on the task-irrelevant background parts.

In this paper, we demonstrated that it is important for the RL model not to be disturbed by task-irrelevant backgrounds for better policy generalization. To do this, we assumed that the RL agent should at least focus more on the task object than on the background, and employed the attribution masks to analyze whether the trained RL models correctly distinguish the task objects and backgrounds. Additionally, to quantitatively analyze how well RL models can distinguish between the task object and the background, we proposed the TID metrics.

In practice, RL policies may not make decisions based on the entire object. Instead, they could care about a few key points within such objects that are most predictive of the reward. To investigate whether the ability to distinguish between task objects and backgrounds also helps in identifying the key points that influence decisions, we provide attention maps obtained from the softmax of the attribution maps $M(Q_\theta, s_t, a_t) = \frac{\partial Q_\theta(s_t, a_t)}{\partial s_t}$ before thresholding for the binarized masks. Fig. 21 shows that the identification ability for the task objects of SimGRL also leads to focus on the most important key point parts such as the pole and body of the 'Cartpole' or legs of the 'Cheetah' in the attention maps. On the other hand, the SVEA baseline incorrectly considers the background features as key points.

# E    Comparison to Robust RL Algorithms

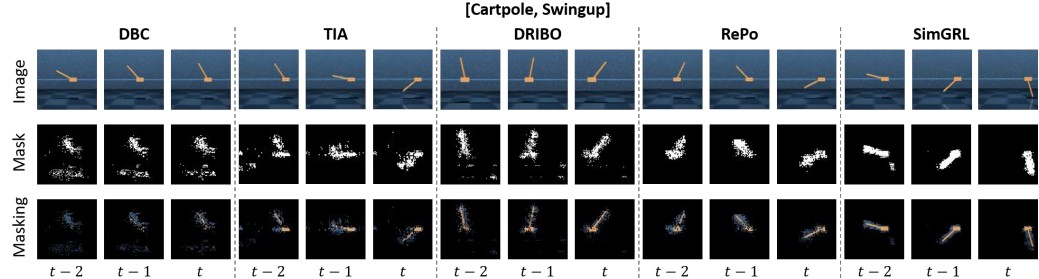

Figure 22: The masking results indicate that DBC, which uses the image-level frame stack, struggles with the imbalanced saliency problem whereas the TIA, DRIBO, and RePo, which feed single images to a recurrent state-space model (RSSM) [10, 9] encoders sequentially, can distinctly identify salient pixels of each frame.

In response to the recommendation of the reviewers to investigate whether the pitfalls also apply to robust RL algorithms such as DBC [43], TIA [7], DRIBO [6], and RePo [47], we provide comparative experimental results for these methods. While our work focuses on addressing the generalization issue in a purely *model-free* vision-based RL setting, which requires the frame stack to encode temporal information, the robust RL approaches aim to explicitly learn robust representations against background distractions by utilizing *model-based* RL. To encode temporal information, these approaches, except for DBC, train transition dynamics by leveraging complex recurrent neural network (RNN) encoders which take only a single frame as input to the encoder, thus applying a concept similar to feature-level frame stacking. On the other hand, DBC uses a CNN encoder that takes stacked frames and trains a dynamics model that predicts the latent states of the next stacked frames. Fig. 22 shows that methods like TIA, DRIBO, and RePo, which use RNN-based encoders, do not struggle with the imbalanced saliency problem, successfully identifying each salient object across consecutive frames like SimGRL. In contrast, despite employing robust representation learning, DBC still struggles with the imbalanced saliency problem, showing a bias toward identifying pixels from the object in the most recent frame as salient across consecutive frames.

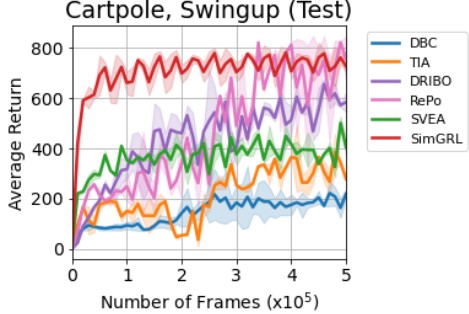

Figure 23: Training curves of DBC, TIA, DRIBO, RePo, SVEA, and SimGRL. In the test stage, we used the video-hard version for the test environments. Compared to robust RL approaches that employ model-based representation learning strategies, the proposed SimGRL achieves both sample efficiency and test performance even with the simple structure.

Fig. 23 shows the curves of zero-shot test performances for the 'Cartpole, Swingup' task in the video-hard environment according to the number of training frames. For a fair comparison, we applied the random overlay augmentation to the robust RL algorithms where following to [13], the data augmentation was applied only during the representation learning stage. As anticipated, robust RL algorithms like DRIBO and RePo, which can distinctly identify task-relevant objects, achieved better test performance (except for TIA) than methods that struggle with imbalanced saliency, such as SVEA and DBC. We suggest that the reason for the inferior performance of TIA is that the TIA agent struggles to train the Distractor Model, which explicitly captures the background, due to the more diverse background augmentations compared to the relatively limited video backgrounds from the Kinetics dataset [16] used in its original experimental setup. Notably, despite its straightforward structure and absence of additional representation learning stages, the proposed SimGRL achieved both sample efficiency and strong test performance, outperforming robust RL approaches like DRIBO and RePo that rely on model-based representation learning strategies.

# F    Details on TID Metrics

## F.1    Dataset for TID Evaluation

To fairly compare TID metrics of comparison methods on the same images, we constructed a dataset for the evaluation in advance. As illustrated in Fig. 24, we excluded samples that were excessively static because the side effect of imbalanced saliency is diminished in such samples. Additionally, we excluded samples where the task object was not fully visible. Instead, we included dynamic samples to fully evaluate the impact of imbalanced saliency. To this end, as the samples obtained from random actions were less dynamic, we selected the images by running SVEA's policies, our baseline algorithm, up to 100K steps to obtain a variety of dynamic samples. Using those images, as depicted in Fig. 25, we constructed 60 pairs of images and ground truth (GT) masks of task objects for each training and testing environment in Video Hard. These GT masks are employed to count $N_{obj}$ and $N_{obj_M}$ in Eq. (8). Here, $N_{obj}$ represents the pixel number in the GT mask while $N_{obj_M}$ is determined by the pixel number in the overlapping regions between the GT and attribution masks.

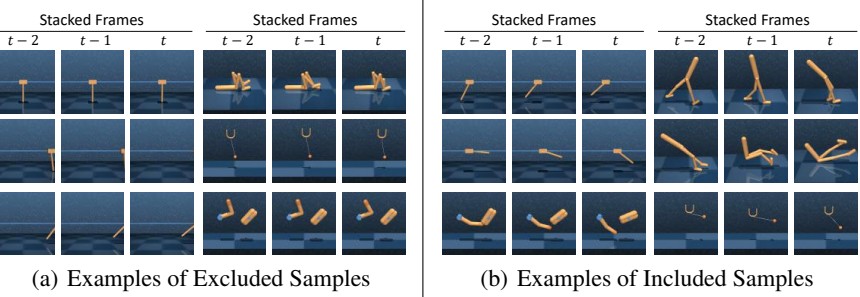

(a) Examples of Excluded Samples    (b) Examples of Included Samples

Figure 24: To evaluate the TID metrics on suitable samples, we excluded overly static or partially visible images for task objects.

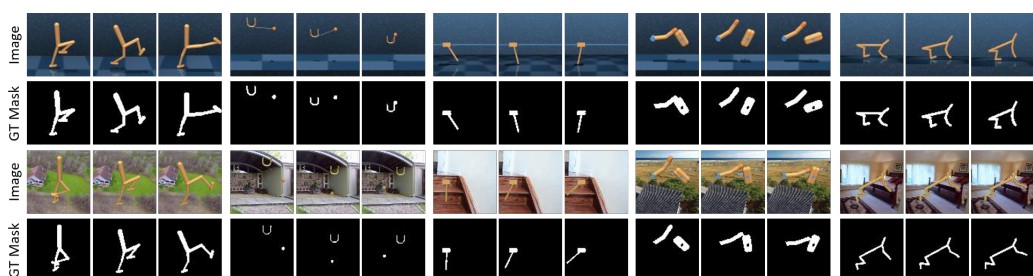

Figure 25: Examples in a dataset for the TID evaluation. (First and Second Row) Images from training environments. (Third and Fourth Row) Images from test environments in Video Hard.

## F.2 Impact of Quantile Value

In Fig. 26, the impact of quantile $\rho$ values for the $\rho$-quantile attribution mask [2] is illustrated. Additionally, the effect of the $\rho$ value on the TID score can be explained as follows. When the value of $\rho$ is too small, the region of a mask is enlarged, thus making it easier for task objects to be included in the attribution mask. This corresponds to the increased $N_{obj_M}$, leading to an increased $\frac{N_{obj_M}}{N_{obj}}$ in the first component of the TID score in Eq. (8). Simultaneously, this also increases $N_M$, balancing the TID score by decreasing $\frac{N_{obj_M}}{N_M}$ that is the second component of the TID score. Conversely, a large value of $\rho$ reduces the size of the mask $N_M$ as in Fig. 26, consequently reducing $N_{obj_M}$. On the other hand, the decreased $N_M$ might increase $\frac{N_{obj_M}}{N_M}$, balancing the TID score. Therefore, for the quantile value used to evaluate the TID score and variance, we used the optimal quantile value computed by $\rho = 1 - \frac{N_{obj}}{(3 \times C \times H \times W)}$ rather than a hand-designed value. However, we used $\rho = 0.95$ in our qualitative illustrations of the attribution masks, as it empirically provided the most plausible visual representation of a masked region.

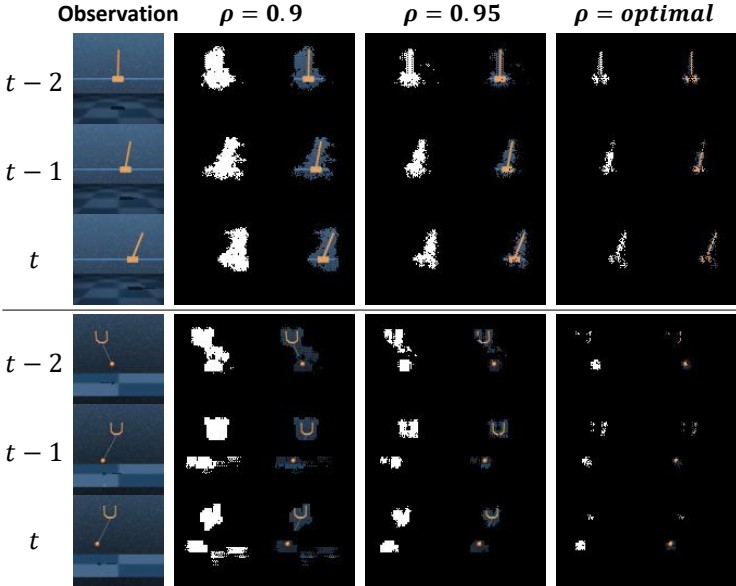

Figure 26: Impacts of quantile values $\rho$ used for thresholding.

## F.3 Full Results of TID Evaluation

We present comprehensive results of the TID evaluation, including a comparison of training TID variance, test TID score, and generalization performance against the training TID score. For the 'Walker' agent, we evaluated the TID scores on the 'Walk' task rather than 'Stand'. Fig. 27(a) reports higher TID scores and lower TID variances of our approach, indicating that SimGRL can distinctly identify the salient pixels across the stacked frames for all tasks. In particular, for the 'Ball In Cup' agent, measuring the TID scores accurately is difficult due to the relatively small size of the salient object. Nevertheless, SimGRL shows remarkably higher TID with lower variance scores than the competitors. These results demonstrate that SimGRL can effectively alleviate the two highlighted issues, 'imbalanced saliency' and 'observational overfitting'. In Fig. 27(b), we can observe that the high identification ability of the task object in the training environment can lead to a similar capability in test environments. Finally, Fig. 27(c) shows a positive correlation between generalization performance and the TID score, where smaller generalization gaps correspond to higher generalization performance. For the distributions in Figures 27(b) and 27(c), we present Pearson correlation coefficients [26] in Table 7. These results imply that the high identification

capability for task-relevant objects across stacked frames in input, as represented by the high TID score, can contribute to improving generalization performance.

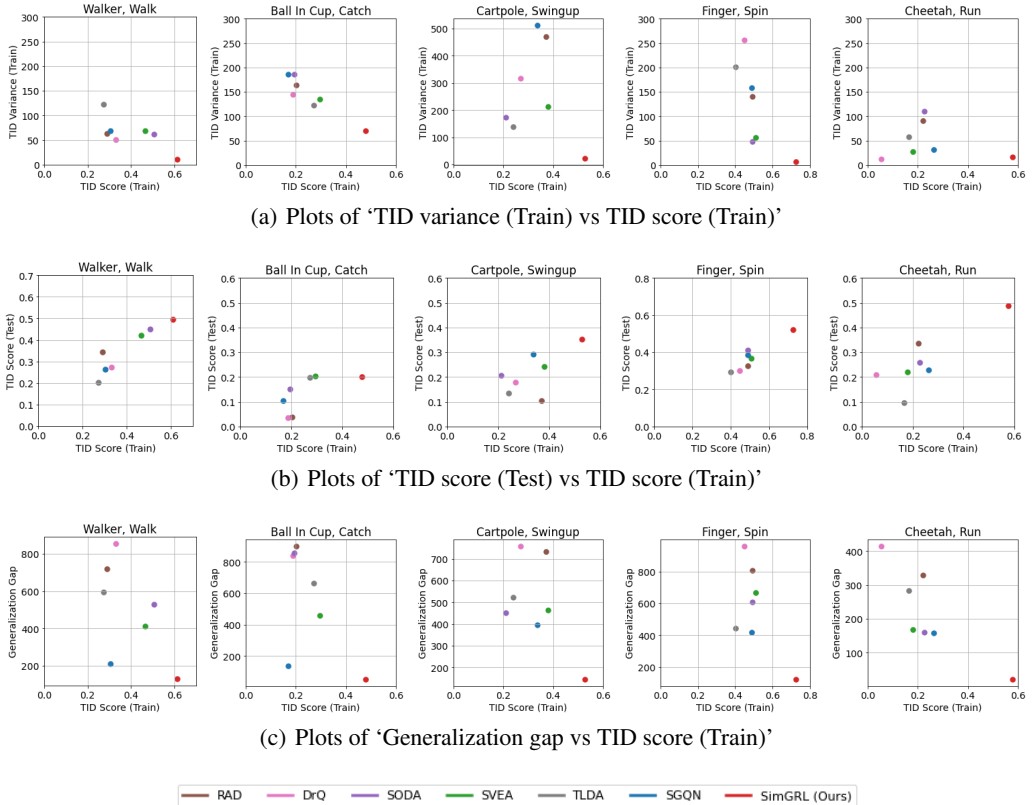

(a) Plots of 'TID variance (Train) vs TID score (Train)'

(b) Plots of 'TID score (Test) vs TID score (Train)'

(c) Plots of 'Generalization gap vs TID score (Train)'

Figure 27: Full results on the TID evaluation.

Table 7: Pearson correlation coefficients.

| Correlation Coefficient | Walker, Walk | Ball In Cup, Catch | Cartpole, Swingup | Finger, Spin | Cheetah, Run | Avg. |
|---|---|---|---|---|---|---|
| $\rho(TID_S^{Train}, TID_S^{Test})$ | 0.92 | 0.66 | 0.62 | 0.93 | 0.83 | 0.79 |
| $\rho(TID_S^{Train}, GenGap)$ | -0.56 | -0.59 | -0.56 | -0.64 | -0.84 | -0.64 |

## F.4 Further Examples of Attribution Masking

For the existing state-of-the-art visual RL approaches for generalization, we provide rich examples of the attribution masks and the corresponding masked observations, where we used the images employed in the TID evaluations. Additionally, we used the quantile $\rho = 0.95$ across all examples. Figures 28 and 29 depict the examples for training and Video Hard testing environments, respectively.

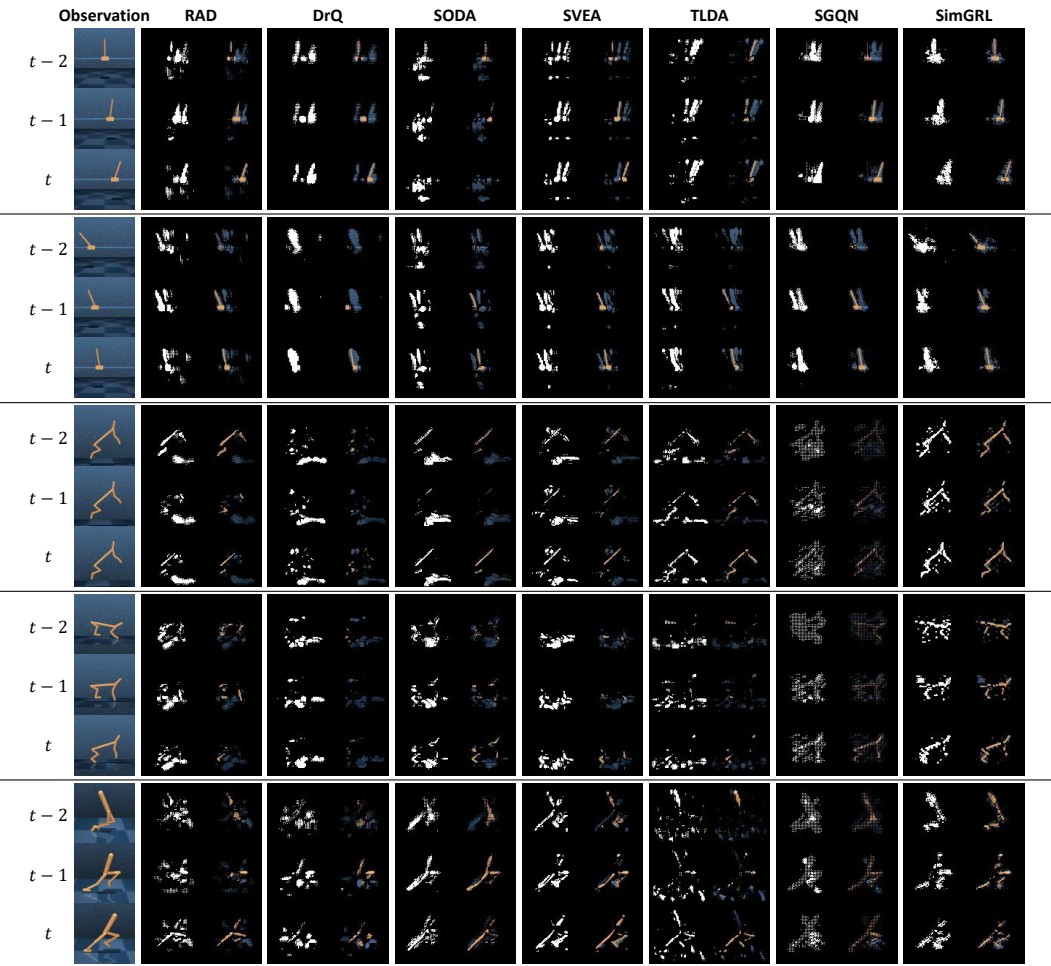

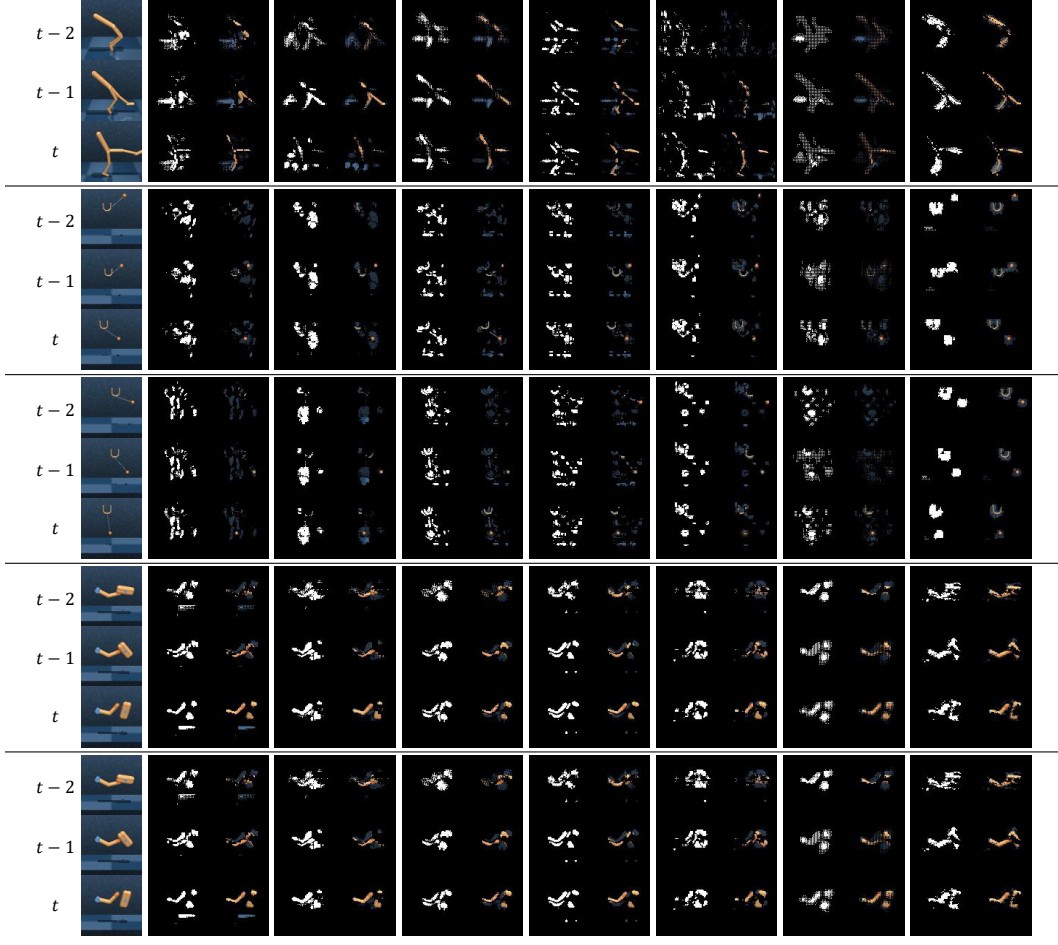

Figure 28: Examples of the attribution masks and masked results for given observations in the training environments.

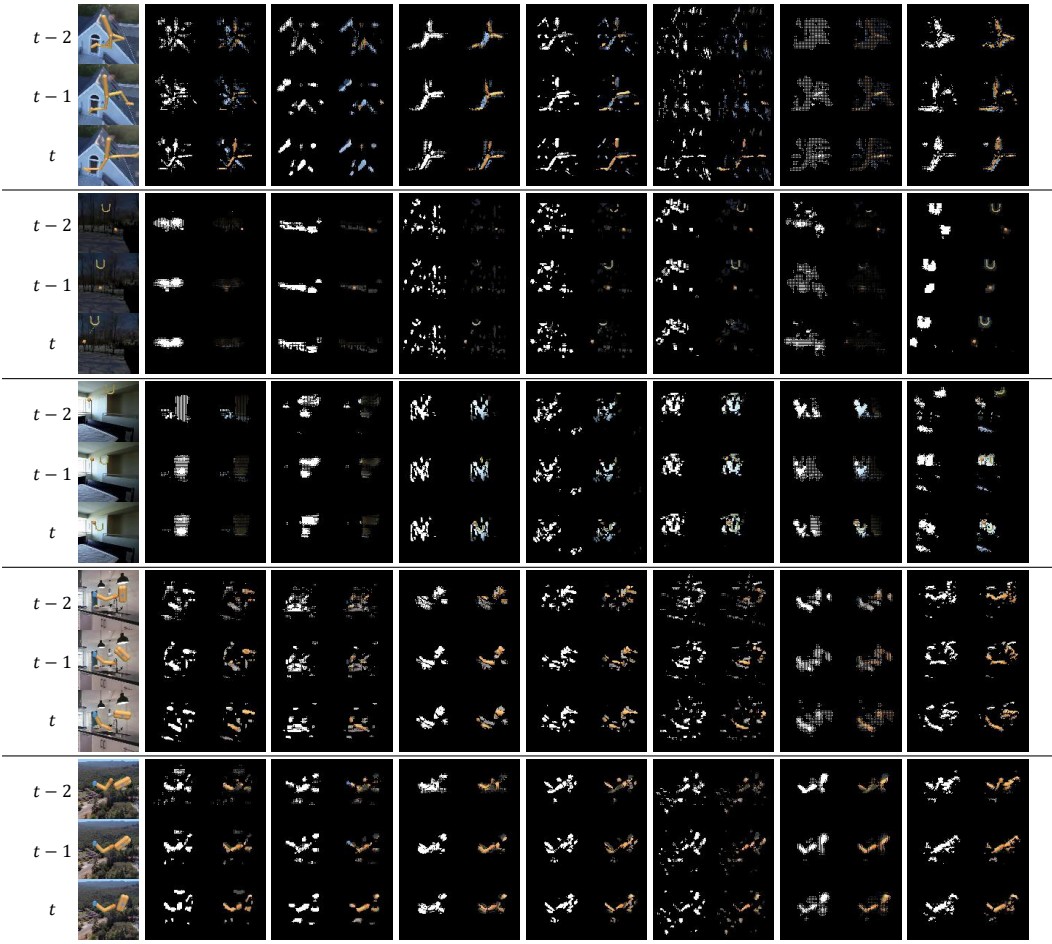

Figure 29: Examples of the attribution masks and masked results for given observations in the test environments of Video Hard.

