# OpenReview forum: "A Simple Framework for Generalization in Visual RL under Dynamic Scene Perturbations"
_NeurIPS.cc/2024/Conference — NeurIPS 2024 poster_

### Official Review · Reviewer_t66L · 2024-07-04

**Soundness:** 3
**Presentation:** 2
**Contribution:** 3
**Rating:** 6
**Confidence:** 4

**Summary:**

This paper introduces SimGRL for vision-based DRL tasks. It tackles two challenges: the imbalanced saliency, where an agent repeatedly favors saliency maps from the recent one in stacked frames, and the issue of overfitting to particular background areas rather than concentrating on elements crucial to the task at hand. To overcome the first challenge, the proposed solution modifies the image encoding process, treating each frame separately before combining their features. To deal with the second issue of observational overfitting, this research uses an innovative method based on shifted random overlay augmentation. This method trains the agent to associate rewards with changes in the actual object of interest, thus learning to ignore background motions. Experiments were carried out using the DMControl-GB video benchmarks, evaluated by the TID score and variance metrics. The results show that SimGRL outperforms existing methods and the ablation study reveals each regularization technique independently contributes to notable enhancements.

**Strengths:**

In this study, the focus is set on addressing critical obstacles in DRL and the author suggested both straightforward and impactful alteration to the model's structure as well as enhancement in the augmentation techniques. The empirical outcomes are reasonable and the statistical findings detailed in Table 1 provide persuasive evidence of the effectiveness of their method.

**Weaknesses:**

The paper could benefit from an explicit articulation of its motivation and contributions. While multiple mentions are naturally made where the two challenges are being addressed, situating the motivation behind the work and a summary of contributions in a dedicated paragraph would enhance the strength of the paper. Although the devised SimGRL algorithm achieves top-tier performance on the Video Hard benchmark, the analysis of its performance on the Video Easy benchmark is not explored in depth, which presents an area for more detailed examination.

**Questions:**

1) The findings concerning Video Easy presented in Table 1 appear to be of lesser importance. Could you provide a more detailed explanation of these findings?

2) Furthermore, Table 1 reveals that 'Cartpole, Swingup' and 'Cheetah, Run' exhibit noteworthy outcomes, although the significance is not observed across all tasks. While you have offered explanations for the strong performance in 'Cartpole, Swingup' and 'Cheetah, Run', can you clarify why there weren't substantial improvements in some other tasks? Does this indicate that the impact of your study might be constrained?

**Limitations:**

No. The paper has limitations, but those are not discussed in the paper. The societal impact is not applicable.

---

> ### Author Rebuttal · Authors · 2024-08-06
>
> We thank the reviewer for your thoughtful comments. We address individual comments in the following.
>
> **[Weakness]**
>
> >**[W1] Need for an explicit articulation of its motivation and contributions**
>
> **(A)** We will add a paragraph to clarify the motivation and contribution of our paper.
>
> ***
>
> >**[W2] Lack of in-depth analysis of the 'Video Easy' benchmark**
>
> **(A)** As for the `video easy’ benchmark, it is much more generalizable than the 'Video Hard' setting, as it features simpler backgrounds and scene structures. Therefore, given that the issues in video hard are resolved, we assume that video easy, which shows high test performance, is also addressed and did not analyze it separately. In addition, the performances for Video Easy of existing methods already are sufficiently saturated performance, and thus there is no significant performance difference between our method and existing methods. Therefore, we did not particularly emphasize the performance on Video Easy. The significance of the results for Video Easy in Table 1 is that our method demonstrates consistently excellent performance across both Video Easy and Video Hard environments, without being specifically favored to any particular environment.
>
>
> ***
>
> **[Question]**
>
> >**[Q1] Need for a more detailed explanation of the results of 'Video Easy' in Table 1**
>
> **(A)** As discussed in **W2**, the significance of the results for Video Easy in Table 1 is that our method demonstrates consistently excellent performance across both Video Easy and Video Hard environments, without being specifically favored to any particular environment.
>
> ***
>
> >**[Q2] Does the strong performance in 'Cartpole, Swingup' and 'Cheetah, Run' indicate that the impact of the study might be constrained?**
>
> **(A)** Existing methods show limited performance on 'Cartpole, Swingup' and 'Cheetah, Run' tasks compared to other relatively easy tasks where the performances are achieved to some extent. In contrast, our method provides consistently sufficient performance across all tasks, regardless of the type and difficulty of the task. Namely, while we see relatively greater performance improvements in the previously underperforming 'Cartpole, Swingup' and 'Cheetah, Run' tasks, we also achieve relatively small yet meaningful gains in other tasks. This is confirmed by the TID metric analysis for all tasks presented in Appendix D.3. In conclusion, we would like to emphasize that the impact of our research is not limited.

---

> > ### Comment · Reviewer_t66L · 2024-08-12
> > **Thank you**
> >
> > I thank the author for answering my questions. The score remains unchanged.

---

> ### Author Response · Authors · 2024-08-13
> **Thank you for your positive comment!**
>
> Dear Reviewer t66L,
>
> We sincerely appreciate your thoughtful response and the time you've dedicated to reviewing our paper.
> We are strongly encouraged by your recognition of our work.
> Thank you once again for your thorough review and your positive evaluation.
>
> Sincerely,
>
> Authors

---

### Official Review · Reviewer_wvnF · 2024-07-09

**Soundness:** 3
**Presentation:** 3
**Contribution:** 3
**Rating:** 6
**Confidence:** 5

**Summary:**

This paper proposes two simple, yet crucial modifications to the Generalization in the Visual RL pipeline where during the test time, the visual observation consists of various degrees of dynamically varying backgrounds.

Concretely, the authors identify two issues that inhibit RL agents from generalization: (i) imbalanced saliency and (ii) observational overfitting. To overcome these issues they propose two simple, yet effective solutions -- (a) Feature level frame stack where instead of stacking frames at the observation level, they stack at a feature level after obtaining spatial features of the images and (b) Shifted Random Overlay Augmentation where instead of applying data augmentation to the stack of frames, a shifted version of the data augmentation is applied to simulate the dynamically moving background effect which in turn helps learn more generalizable policy.

Through their experiments on the DM-Control (and robotic manipulation in the appendix) with varying backgrounds, the authors show the generalization capability of their proposed SimGRL (Simple framework for Generalization in visual RL) framework.

Additionally, the authors also propose Task-Identification (TID) Metrics to evaluate how much of the policy can capture the task-relevant objects in each stacked frame.

**Strengths:**

1. A key strength of this paper is the simplicity of the proposed solutions to address the generalization issue and these simple changes are quite effective as shown in the experiments.

2. The intuition behind both the proposed solutions is clear and straightforward.

**Weaknesses:**

1. I do find myself in confusion with respect to the Attribution masking argument. I believe that the proposed solutions of Feature-level frame stack and Shifted Random overlay augmentation are both valid solutions to a generalizable policy irrespective of the authors inclusion of Attribution masking. My specific concern is as follows:

    What is the guarantee that a clear thresholded gradient masking of the critic wrt the input, which consists of the complete segmentation of the relevant objects and agent in the mask will lead to a better generalizable policy? It can be possible that a generalizable policy would care about only a few key points that are predictive of the reward -- and hence be able to accurately predict the Value of a state, and in that sense, an accurate segmentation of the attribution mask need not be necessary. To give an example, in the robotic manipulation task of reach -- the reward function is typically shaped as some function of the Euclidean distance between the robot's end-effector and the goal position. In such a situation, the Value function shouldn't really be bothered about where the other parts of the robot are.

2. The weakness in (1) ties into my second concern where if only a few parts of the agent are responsible for achieving (and predicting) reward, then I don't see the value of the proposed TID metric. This assumes that my entire object and agent in the scene is responsible for the policy, which as argued in (1) is not the case in most scenarios.

3. [Relatively minor concern, not taken into account for the final scoring] It would be beneficial to the reader if the proposed components in Figure 4 could be highlighted with say a different color and a distinction between the existing pipeline of SVEA be made.

4. On the experimentation end, I feel that the evaluation is very limited to DM-control tasks. While SimGRL performs quite well on Video-Easy and Video-Hard settings of DM-Control, and 2 other tasks of Robotic manipulation, I would have liked to see a more comprehensive evaluation of various visual RL environments. Specifically, I'd recommend performing evaluation on RL-ViGen [1] (https://gemcollector.github.io/RL-ViGen/)  which consists of a collection of diverse visual RL tasks. This would further strengthen this work's results.

5. On L263, where authors say:
> SimGRL demonstrates state-of-the-art performance in 4 out of 6 tasks in the Video Easy benchmark

    One has to be careful while making such claims especially when the difference on one such task is +1 -- which in DM-Control doesn't really mean anything. I would ask the authors to look at IQM and Probability of Improvement metrics to compare SimGRL to other methods from [2] which has shown to be more reliable than taking an average over seeds. This does not involve re-running the experiments, the existing checkpoints and returns can be directly used over their wrapper.

**Questions:**

6.  Observational Overfitting: For this, one of the core arguments that authors provide is that because the same augmentation is applied uniformly to all frames the agent can tend to still focus on the background. I'm wondering how a baseline for this where you have different data augmentations for every frame in the history of stacked frames(say jittering, slight rotation, blur, overlay, random convolution, random shift) performs?

7. In the Robotic manipulation results what does Test {1, 2, 3} mean? Does it mean running an evaluation on a _single episode_ for 5 seeds?

----
**References**

[1] RL-ViGen: A Reinforcement Learning Benchmark for Visual Generalization, Zhecheng Yuan et al., NeurIPS 2023.

[2] Deep Reinforcement Learning at the Edge of the Statistical Precipice, Rishabh Agarwal et al., NeurIPS 2021.

----
**Reason for current rating**: Because of my two major concerns around Attribute masking (and TID metrics), as well as lack of extensive experimentation on diverse RL environments, I am leaning towards a weak reject. However, my final decision would be based on authors' rebuttal, other reviewers comments and the discussion with authors during the rebuttal period. I'd encourage the authors to ask me any clarification questions for any of the experiments if they have any during the rebuttal/discussion period.

-----

**Post rebuttal update** -- As the authors have addressed all my major concerns, I have decided to increase my score to **Weak Accept**.

**Limitations:**

The authors don't mention any limitations of their work. It would be good to have a couple of lines highlighting where their method may fail. For instance, potentially large changes in background can lead to failure or their Feature stacking would work only in situations where Frame stacking is used -- model-based works like Dreamer do not use any frame stacking because of a recurrent network (such as GRU) that handles the history of states.

---

> ### Author Rebuttal · Authors · 2024-08-06
>
> We thank the reviewer for your thoughtful comments. We address individual comments in the following. Additionally, we sincerely appreciate your eagerness to engage in further discussions regarding our work!
>
> **[Weakness]**
>
> >**[W1] Clarity w.r.t the attribution masking argument**
>
> **(A)** *To avoid confusion, we clarify that our paper does not claim that an RL model must make a decision based on the ‘whole’ of task-relevant objects for better policy generalization.  Rather, we emphasize that it is important for the RL model not to be disturbed by task-irrelevant backgrounds.* To do this, we assumed that the RL agent should at least focus more on the task object than on the background, and employed the attribution masks to analyze whether the trained RL models correctly distinguish the task objects and backgrounds. Additionally, to quantitatively measure and analyze the ability to distinguish, we proposed the TID metric. We clarify that this metric is not designed to measure whether the models are making decisions based on the entire object.
>
> To investigate whether the ability to distinguish between task objects and backgrounds also helps in identifying the key points that influence decisions, we provide attention maps obtained from the softmax of the attribution maps $M(Q_\theta, s_t, a_t)$ before thresholding for the masks $M_\rho(Q_\theta, s_t, a_t)$ in the attached PDF. Figure 3 of the PDF shows that the identification ability for the task objects of SimGRL also leads to focus on the most important key point parts such as the pole and body of the 'Cartpole’ or legs of the 'Cheetah’ in the attention maps. On the other hand, the SVEA baseline incorrectly considers the background parts as key points.
>
> ***
>
> >**[W2] TID Metric when a few parts of the agent are responsible for achieving reward**
>
> **(A)** As addressed in **W1**, we do not assume that the entire object is responsible for the policy's decision; instead, we assume it is important for the agent not to be disturbed by task-irrelevant backgrounds. Therefore, TID metrics are designed to measure how well the task-relevant object and background are distinguished, without considering which parts within the task-relevant object are more important.
>
> ***
>
> >**[W3] Need for improvement of Figure 4**
>
> **(A)** We will improve Figure 4 by highlighting a distinction between the existing pipeline of the SVEA baseline. The modified figure is presented in Figure 2(b) in the PDF.
>
> ***
>
> >**[W4] Additional experiments for RL-ViGen benchmark**
>
> **(A)** Due to the time constraints of the rebuttal period, we were only able to conduct a limited number of additional experiments on the RL-ViGen benchmark task.
> Specifically, we present the experimental results on the 'Door' task in the Robosuite environment.
>
> |Task : Door|	SVEA|	SimGRL|
> |-|-|-|
> |**Train**|	477.60|	482.58|
> |**Test (Easy)**|	112.86|	200.51|
>
> This table demonstrates that SimGRL achieves test performance gain against the baseline method in different simulation environments as well.
>
> ***
>
> >**[W5] Measurement using IQM metric**
>
> **(A)** We evaluated the IQM (interquartile mean) scores, where we discarded the top and bottom values over the 5 seeds, and averaged the middle 3 seeds.
>
> ||DMC-GB|SAC|RAD|DrQ|SODA|SVEA|TLDA|SGQN|SimGRL|
> |-|-|-|-|-|-|-|-|-|-|
> |**Video Easy**|Walker, Walk|262|540|741|	846|	888|	847|	895|	**905**|
> ||Walker, Stand|322|	898|	954|	953|	955|	950|	962|	**966**|
> ||Ball In Cup, Catch|	134|	432|	495|	658|	922|	868|	869|	**964**|
> ||Cartpole, Swingup|	348|	478|	428|	705|	779|	695|	642|	**849**|
> ||Finger, Spin|	262|	517|	514|	687|	790|	640|	920|	**982**|
> ||Cheetah, Run|56|	114|	251|	243|	237|	271|	300|	**311**|
> |**Video Hard**|Walker, Walk|	71|	54|	88|	378|	532|	424|	726|	**769**|
> ||Walker, Stand	|198|	240|	311|	792|	845|	650|	872|	**934**|
> ||Ball In Cup, Catch|	85|	73|	131|	116|	514|	472|	835|	**902**|
> ||Cartpole, Swingup|	112|	135|	120|	410|	405|	320|	383|	**723**|
> ||Finger, Spin|	12|	82|	24|	313|	311|	230|	567|	**868**|
> ||Cheetah, Run|	23|	15|	38|	158|	99|	91|	171|	**299**|
>
> The table shows that SimGRL also achieved robust performance in IQM evaluation.
>
> ***
>
> **[Question]**
>
> >**[Q1] How does a baseline where we have different data augmentations for every frame in the history of stacked frames perform?**
>
> **(A)** To augment the background, it is better to apply the augmentation non-uniformly. However, if the augmentation directly affects the agent, using uniform augmentation would be more appropriate. For example, if random shifts are applied differently, the agent's position may change for each frame, which may distort temporal information. Therefore, the random shift (referred to as soft augmentation in the paper) applied by default, like in existing methods, was applied uniformly. On the other hand, we confirmed that shifting the random overlay, which has a greater influence on the background than the agent, is effective for generalizing to environments including dynamic perturbations.
>
> ***
>
> >**[Q2] In the Robotic manipulation results what does Test {1, 2, 3} mean? Does it mean running an evaluation on a single episode for 5 seeds?**
>
> **(A)** Robot manipulation experiments were conducted by testing in each of the test environments—test 1, test 2, and test 3—after training in the training environment. Each test environment was evaluated with 30 episodes, and this process was repeated across a total of 5 different seeds for training.

---

> > ### Comment · Reviewer_wvnF · 2024-08-12
> > **Thanks for addressing my major concerns and for the clarification**
> >
> > Thanks to the authors for their rebuttal.
> >
> > 1. The clarification on the attribution masking is now helpful for me to understand the claims they are making and helped clear my confusion. I'd request the authors to add maybe a sentence or two in the main paper as well (either for camera ready if accepted or for future submissions).
> >
> > 2. Thanks for showing the IQM over 5 runs.
> >
> > 3. Re: Q1 -- I was hoping for an empirical evidence but I do get the authors' point. By _random augmentation_, I was implying the typical computer vision augmentation (color jitter, crop, rotate etc) but independently for each frame in the stack rather than applying the same augmentation for the entire 3 frames. But in hindsight looking back at this comment I do think that even the random shift would be better as that is a stronger form of data augmentation.
> >
> > 4. Thanks for the clarification on the test1, 2, 3 and for mentioning the number of episodes on which the models were evaluated.
> >
> > Based on the rebuttal, I'm convinced that this paper is in a good quality for publication and hence am increasing my score to Weak Accept. Once again, thank you to the authors for their rebuttal.

---

> ### Author Response · Authors · 2024-08-13
> **Thank you for your positive comment!**
>
> Dear Reviewer wvnF,
>
> We sincerely appreciate your thoughtful response and the time you've dedicated to reviewing our paper.
> We are strongly encouraged by your recognition of our work. We will incorporate your suggestions and insights into the revised manuscript.
> Specifically, we will add some sentences to avoid confusion and enhance the clarity of our claims.
> Thank you once again for your thorough review and your positive evaluation.
>
> Sincerely,
>
> Authors

---

### Official Review · Reviewer_kgwp · 2024-07-13

**Soundness:** 3
**Presentation:** 3
**Contribution:** 2
**Rating:** 6
**Confidence:** 4

**Summary:**

The paper first identify two pitfalls with the visual RL via gradient-based attribution mask, i.e., imbalanced saliency and observational overfitting. To address these two pitfalls, the paper proposes two novel modifications. One is for the encoder where encode each frame and stack encoded representations instead of encoding stacked frames directly. The other is a newly proposed shifted  randomly overlay augmentation method to address the observational overfitting.

**Strengths:**

- By and large, the paper is written well and presented clearly.
- The section on pitfalls within visual RL algorithms is also interesting, and highlights the authors attention to the RL robustness and generalization problems.

**Weaknesses:**

- One concern I have about the paper is the first pitfall where the imbalanced saliency is specifically applied to visual RL algorithms that apply stacked frames. However, many robust RL algorithms, e.g., DBC, SLAC, TiA, DRIBO, RePo, take only single frame as the input to the encoder which apply similar ideas as the feature-level frame stack. These algorithms also explicitly take previous frame stack as input with an encoder $p_\theta(s_t | s_{t-1}, a_{t-1}, o_t)$. It would be interesting to investigate whether these methods also suffer from the imbalanced saliency and compare the proposed feature-level stack method with them in terms of performance and efficiency.
- Though the paper identifies the pitfalls with attribution methods, it would also be interesting to see attribution masks of SimGRL in different settings, i.e., clean, video easy and video hard. This would help validate that the identified pitfalls are sufficiently addressed by SimGRL.
- Fig 5 and demos showed along the paper may be misleading that the policy is purely learned from the clean environment without any background perturbation. However, the proposed randomly shifted overlay augmentation explicitly introduces background perturbations into the training process.

**Questions:**

- Could you comment on whether the pitfalls also apply to robust RL algorithms like DBC, SLAC, TiA, DRIBO and RePo?
- Could you comment on how different attribution methods may affect the identified pitfalls and TID metrics?

**Limitations:**

Yes, the author addressed the limitations in the paper.

---

> ### Author Rebuttal · Authors · 2024-08-06
>
> We thank the reviewer for your thoughtful comments. We address individual comments in the following.
>
> **[Weakness]**
> >**[W1] Comparison with robust RL algorithms**
>
> **(A)** We clarify that our work aims to address the generalization issue in a purely **model-free vision-based RL** setting, which requires the frame stack to encode temporal information. Due to the limitations of the rebuttal period, we present comparative results for DBC[1] and DRIBO[2] among the robust RL algorithms to investigate whether these methods also suffer from imbalanced saliency and compare them with the proposed method.
>
> + DBC: Image-level frame stacking of 3 images, CNN encoder, Representation learning using a dynamics model, No strong data augmentation (e.g., Random overlay).
>
> + DRIBO: No frame stack (using single images), Recurrent State-Space Model (RSSM) encoder, Representation learning using a dynamics model, No strong data augmentation (e.g., Random overlay).
>
> + SimGRL (Ours): Feature-level frame stack of 3 images, CNN encoder, No representation learning and a dynamics model, Strong data augmentation (i.e., Shifted Random overlay).
>
> To investigate whether these methods suffer from imbalanced saliency, we investigated TID metrics on 'Cartpole, Swingup', and compared them with SimGRL
> ||DBC|	DRIBO|	SimGRL|
> |-|-|-|-|
> |**TID Score**|	0.375|	0.483|	0.528|
> |**TID Variance**|	284.5|	33.7|	22.8|
>
> This table indicates that *1) DBC suffers from imbalanced saliency* (a lower TID Score and higher TID Variance) and *2) DRIBO does not suffer from imbalanced saliency* and can correctly recognize individual images (a higher TID Score and lower TID variance). Additionally, for better understanding, we provide the qualitative results of the masks and masked images for DBC and DRIBO in Fig. 1(a) of the attached PDF. These results imply that, when using an image-level frame stack as in DBC, imbalanced saliency persists even when the model-based RL setting is integrated. In contrast, when encoding single frames individually as in DRIBO, imbalanced saliency can be avoided, similar to feature-level frame stacking of the proposed method.
>
> However, our SimGRL has the advantage of being a simple yet effective structure, as it uses neither complex RNN-based networks like RSSM nor any additional representation learning strategies. In Figure 1(b) of the attached PDF, the training curves indicate the high sample efficiency in the training stage and superior test time performance of SimGRL.
>
> ***
>
> >**[W2] Discussion on attribution masks in different settings**
>
> **(A)** In Section 5.2 and Appendix D.3, we have conducted an in-depth analysis of the pitfalls regarding clean videos and video hard scenarios. Additionally, in Appendix D.4, we have included rich example images of various attribution masks to demonstrate how effectively SimGRL addresses these pitfalls compared to existing methods. We would like to request you to refer to these sections. As for the 'Video Easy’ setting, it is much more generalizable than the 'Video Hard' setting, as it features simpler backgrounds and scene structures. Therefore, considering that the issues in the 'Video Hard’ setting are resolved, we assume that in the `Video Easy’ setting, which shows high test performance, the attribution mask is also well predicted, so we did not analyze it separately.
>
> ***
>
> >**[W3]  Figure 5 that may be misleading**
>
> **(A)** For better clarity, we will add the augmentations used during training in Figure 5. We provide the modified figure in Figure 2(a) in the attached PDF.
>
> ***
>
> **[Question]**
>
> >**[Q1] Whether the pitfalls also apply to robust RL algorithms**
>
> **(A)** As explained in **W1**, even robust RL algorithms can experience pitfalls when using image-level frame stacks, as seen with DBC, but these pitfalls will be resolved when sequentially embedding single images, as done in DRIBO.
>
> ***
>
> >**[Q2] How different attribution methods may affect the identified pitfalls and TID metrics**
>
> **(A)** We clarify that the attribution masks are used solely for analysis and not for training, so they do not impact whether the RL models fall into pitfalls during training. Therefore, even if different methods are used to obtain attribution masks, as long as they correctly measure the influence of pixels in input images on the RL model, the identified pitfalls and TID metric values will not change significantly.
>
>  ***
> [1] "Learning invariant representations for reinforcement learning without reconstruction." ICLR (2021).
>
> [2] "Dribo: Robust deep reinforcement learning via multi-view information bottleneck." ICML (2022).

---

> > ### Comment · Reviewer_kgwp · 2024-08-10
> > **Thank you for your detailed response!**
> >
> > I would like to first thank authors for the detailed response and new comparison results. These are super helpful.
> >
> > Most  of my concerns have been addressed except W1 and Q1. I would love to raise my score if authors can help better understand W1 and Q1 with the following questions.
> >
> > ## W1 and Q1
> >
> > Thank you for providing the comparison results with DBC and DRIBO. These results are very helpful.
> >
> > - A clarification question, are DBC and DRIBO trained with the same video augmentation as SimGL or trained in the clean setting? I am asking because Fig. 1(b) in the response letter showed that both DBC and DRIBO does not perform well during testing. I am wondering if author have any comments on it.
> >
> > - The new results addressed my concern of whether existing robust RL methods addressed imbalanced saliency issue. Since the other major contribution of the paper is the random overlay augmentation, I am wondering if existing robust RL methods can also benefit from the random overlay augmentation.

---

> ### Author Response · Authors · 2024-08-13
>
> We thank you for your constructive feedback and your eagerness to engage in further discussions regarding our work!
>
> We address your comments in the following.
>
> ***
>
> > **[Q1] Are DBC and DRIBO trained with the same video augmentation as SimGL or trained in the clean setting?**
>
> **(A)** To clarify whether robust RL methods suffer from the imbalanced saliency problem, and since both DRIBO and DBC did not originally use strong augmentation such as random overlay, clean images were used during training.
> This implies that even when methods like DRIBO, which learn robust representations, do not suffer from issues such as imbalanced saliency, training on diverse backgrounds remains important for generalization to unseen environments.
>
> ***
>
> > **[Q2] Can existing robust RL methods also benefit from the random overlay augmentation?**
>
> **(A)**  To investigate whether existing robust RL methods also benefit from augmentations like random overlay, we provide experimental results on the 'Cartpole, Swingup' task in the 'Video Hard' test environment for DBC and DRIBO when using random overlay augmentations.
> Following SODA [1], we applied random overlay augmentations solely to the representation learning losses of the encoder for stability in training.
>
> ||Train	|No Aug.	|Random Overlay	|Shifted Random Overlay|
> |-|-|-|-|-|
> |**DBC**	|805.2	|83.7	|169.7	|170.4|
> |**DRIBO**	|818.6	|134.7	|414.8	|532.3|
>
> The above table indicates that DBC showed marginal effects even with the application of random overlay, while DRIBO demonstrated clear improvements. Additionally, DRIBO was able to obtain additional performance gain when using the shifted random overlay augmentation. For DRIBO, these results clearly suggest that robust RL algorithms can benefit from data augmentation techniques such as random overlay, further strengthening our contribution. On the other hand, DBC's results suggest that a specific augmentation method may not always be well-suited for a particular learning algorithm. Alternatively, this could be due to issues with hyperparameter tuning in methods with complex representation learning processes, where the marginal performance might be attributed to not finding the optimal hyperparameters for applying the augmentation. Our approach, with its simple structure and minimal hyperparameter tuning, could be a solution to address these drawbacks.
>
> ***
> **References**
>
> [1] "Generalization in reinforcement learning by soft data augmentation." ICRA (2021).

---

> ### Comment · Reviewer_kgwp · 2024-08-13
>
> Thank you for the further clarification!
>
> The results on applying random overlay augmentation on DBC and DRIBO are particularly helpful. I would rather disagree with the setting where DBC and DRIBO are compared with SimGL but trained under the clean setting. Both DBC and DRIBO did not claim transferability of robust RL agents trained under the clean setting. I would recommend author to include the second part of results in the revised version.
>
> In addition, I would strongly recommend authors to compare with more recent robust RL methods in the revised version.
>
> I would like to raise my score to 6 and this is a borderline paper to me. I think the major contribution of the paper lies on the random overlay augmentation and the TID metric. However, more comparison with other robust RL methods can further strengthen the paper.

---

> > ### Author Response · Authors · 2024-08-13
> > **Thank you for your positive comment!**
> >
> > Dear Reviewer kgwp,
> >
> > We sincerely appreciate your thoughtful response and your dedicated time to review our paper.
> > We are strongly encouraged by your recognition of our work.
> > We will incorporate your suggestions and insights into the revised manuscript.
> > Specifically, we agree with your comments that DBC and DRIBO need to be compared with SimGRL under the setting using the same data augmentation and we will add a section in the Appendix of the revised version to discuss the robust RL algorithms including the second part of the results of the rebuttal.
> > Additionally, we will make an effort to include results comparing recent robust RL methods, such as TiA [2] and RePo [3], in the revised version.
> > Thank you once again for your thorough review and your positive evaluation.
> >
> > Sincerely,
> >
> > Authors
> >
> > ***
> >
> > **References**
> >
> > [2] "Learning task informed abstractions." ICML (2021).
> >
> > [3] "Repo: Resilient model-based reinforcement learning by regularizing posterior predictability." NeurIPS (2023).

---

### Official Review · Reviewer_z8gS · 2024-07-16

**Soundness:** 3
**Presentation:** 4
**Contribution:** 3
**Rating:** 6
**Confidence:** 3

**Summary:**

The paper introduces SimGRL, a framework aimed at enhancing generalization in vision-based deep reinforcement learning (RL) under dynamic scene perturbations. It addresses two critical issues in existing visual RL methods: imbalanced saliency and observational overfitting. SimGRL employs architectural modifications to the image encoder and a novel shifted random overlay augmentation technique. Extensive experiments demonstrate SimGRL's superior generalization capabilities, achieving state-of-the-art performance on the DeepMind Control Suite and other benchmarks.

**Strengths:**

Innovative Architectural Modification: The proposed modification to stack frames at the feature level instead of the image level effectively addresses the imbalanced saliency issue, ensuring the agent focuses on spatially salient features in each frame .
Novel Data Augmentation Technique: The shifted random overlay augmentation introduces dynamic background elements during training, which helps the agent to focus on task-relevant features and ignore irrelevant background changes .
Comprehensive Evaluation: The method is thoroughly tested on multiple benchmarks, including the DeepMind Control Suite and DMControl-GB, demonstrating superior generalization performance compared to state-of-the-art methods.
Quantitative Metrics: The introduction of TID metrics provides a quantitative way to evaluate and understand the method's effectiveness in identifying task-relevant features, correlating high TID scores with improved generalization performance.
Computational Efficiency: The method achieves significant improvements without requiring additional auxiliary losses or networks, maintaining computational efficiency.

**Weaknesses:**

Dependency on Augmentation Quality: The effectiveness of the shifted random overlay augmentation relies on the quality and diversity of the natural images used for augmentation, which may limit its performance in certain real-world scenarios where such images are not available or appropriate.
Ad Hoc approach: the authors offer specific fixes to observed shortcomings in vision-based deep RL in simulated environments, and it remains to be seen if these fixes are an hand-crafted overfitting to specific tasks that won't generalize to other tasks.
Finally, while the method shows impressive results in simulated environments, its applicability and effectiveness in more realistic real-world scenarios have not been thoroughly validated.

**Questions:**

How does the diversity of natural images used in the shifted random overlay augmentation impact the generalization performance of the RL agents? Are there specific types of images that are more effective?
Are there any plans to test SimGRL in real-world environments, and what challenges are anticipated in such scenarios? How does the method handle real-world dynamic perturbations that are not present in simulated benchmarks?
What are the effects of using different numbers of frames in the feature-level frame stack on the performance and generalization capability of SimGRL?
How sensitive is the performance of SimGRL to the choice of hyperparameters such as the number of layers in the image encoder and the maximum shift length in the shifted random overlay augmentation?

**Limitations:**

The architectural modifications and data augmentations, while effective, still introduce some computational overhead, which may impact training times and computational resources required.
The method is specifically designed for vision-based RL and may not directly apply to other types of RL tasks without visual components.
The shifted random overlay augmentation relies on the quality and diversity of natural images used for augmentation, which may limit its performance in scenarios where such images are not available or appropriate.

---

> ### Author Rebuttal · Authors · 2024-08-06
>
> We thank the reviewer for your thoughtful comments. We address individual comments in the following.
>
> **[Weakness]**
>
> ***
>
> >**[W1] Dependency on augmentation quality**
>
> **(A)**  The effectiveness of the shifted random overlay augmentation may depend on the quality and diversity of the natural images, and the performance may be limited in specific real-world scenarios where such images are unavailable or unsuitable. Nevertheless, we demonstrated in **Q1** that even with fewer and less diverse augmentation data, robust performance can be maintained.
>
> ***
>
> >**[W2] Ad Hoc approach**
>
> **(A)**  SimGRL consistently demonstrates robust generalization performance across various distraction environments and task scenarios, including DMControl-GB (Section 5.1 and Appendix B.6), DistractingCS (Appendix B.7), and robotic manipulation (Appendix B.8). Therefore, we do not consider this to be overfitting to specific tasks.
>
> ***
>
> >**[W3] Applicability to real-world scenarios**
>
> **(A)**
> Because it is very difficult to build a safe real-world RL experimental environment using actual robots, existing studies have primarily verified algorithms in simulation environments. Similarly, we conducted experiments in simulation environments to compare our method with existing methods. Nonetheless, applying the algorithm to more realistic real-world situations is crucial for practical use, and we also consider this an important area for future research.
>
> ***
>
> **[Question]**
>
> >**[Q1] Impact on the diversity of natural images**
>
> **(A)** We provide the average results of SimGRL for the training set and validation set of the Places dataset used for augmentation on the DM-Control-GB benchmark.
>
> ||Training set (1.8M Scenes)|Validation Set (36000 Scenes)|
> |-|----|---|
> |**Reward**|753.17|751.13|
>
> This table implies that even with fewer types of natural images, the performance does not significantly degrade, implying that the diversity of natural images does not have a major impact on generalization performance. More effective types of images for augmentation would be those that resemble the distribution of the test environment. However, since we do not have prior information about the test environment, we used arbitrary natural images for augmentation and confirmed through various experiments that this approach is effective.
>
> ***
>
> >**[Q2] Challenges anticipated in real-world scenarios**
>
> **(A)** A potential challenge when testing in real environments is the distribution shift, where the distribution of natural images used for augmentation during training differs from the distribution in the actual environment. However, our approach trains the model to focus on task-relevant objects while ignoring the background. As these task-relevant objects should consistently exist even in new environments with different backgrounds, we expect SimGRL to perform well in such cases.
>
> ***
>
> >**[Q3] Perturbations that are not present in simulated benchmarks**
>
> **(A)** The robust performance of our method in DistractingCS in Appendix B.7, which includes perturbations such as camera pose distractions, suggests that our approach can handle types of disturbances not present in the training environment. We speculate that SimGRL’s robustness to distractions in this benchmark stems from its excellent ability to identify salient objects in the input, which helps in generalizing against distractions such as dynamic backgrounds or changes in camera pose.
>
> ***
>
> >**[Q4] Effects of numbers of frames in the feature-level frame stack**
>
> **(A)** The number of stacked frames involves a trade-off between the amount of temporal information and the computational cost for encoding. If the number of stacked frames is reduced to less than three, the performance will degrade due to insufficient temporal information. On the other hand, since a feature-level frame stack encodes individual images separately in the image encoder, the number of encodings required increases in proportion to the number of frames, resulting in higher computational costs. Therefore, we adopted stacking three frames as in the previous methods and experimentally confirmed that this is the most appropriate frame stack.
>
> ***
>
> >**[Q5] Sensitivity of the performance to the choice of hyperparameters**
>
> **(A)** As analyzed in Appendix B.3, the sensitivity to the number of layers in the image encoder was not highly critical. Even with just one layer, using a feature-level frame stack demonstrates significantly better generalization performance compared to an image-level frame stack.
>
> Additionally, to show the impact of varying the maximum shift length (L) for the shifted random overlay, we present the test results on the "Cartpole, Swingup" task in 'Video Hard' for SimGRL-S, which uses only shifted random overlay augmentation without a feature-level frame stack.
>
> ||L = 0|L = 3|L = 6|L = 9|
> |-|-|-|-|-|
> |**Reward**|393|713|724|701|
>
> In this case, sensitivity to the maximum shift length is not very high, but there is a significant difference compared to the case of not using any shifting (L=0).

---

### Author Rebuttal · Authors · 2024-08-07

Dear reviewers, we appreciate all your valuable comments and the time you've dedicated to reviewing our paper.

Here, we briefly summarize the answers to some commonly asked questions and upload a PDF file with some additional figures to aid in understanding.

+ The attribution masks were used solely for identifying and analyzing the found pitfalls in existing model-free vision-based RL algorithms—imbalanced saliency and observational overfitting—and were not utilized during training.

+ Our main contributions are: 1) to the best of our knowledge, the first in-depth analysis of the pitfalls that hinder generalization in existing algorithms from the perspective of input frames, and 2) the demonstration that these problems can be alleviated and generalization significantly improved with a few simple modifications—namely, feature-level frame stacking and shifted random overlay augmentation.

+ We provide additional comparisons with methods that integrate model-based approaches, specifically DBC and DRIBO. The results are included in Figure 1 in the attached PDF.


Sincerely,

Authors

---

### Decision · Program_Chairs · 2024-09-25

**Decision:**

Accept (poster)

**Comment:**

The reviewers advocate accepting this paper, and the AC agrees with this evaluation. During the rebuttal phase, the authors clarified the reviewers' confusion on the technical aspects and the experimental results. The reviewers found that the rebuttal adequately resolved the weaknesses. One minor note, the authors should update their references. Specifically, the authors cited the arXiv version instead of the conference version, e.g. [37] has been accepted to ICLR.